# GENERATIVE NEGATIVE REPLAY FOR CONTINUAL LEARNING

## ABSTRACT

Learning continually is a key aspect of intelligence and a necessary ability to solve many real-world problems. One of the most effective strategies to control catastrophic forgetting, the Achilles' heel of continual learning, is storing part of the old data and replay them interleaved with new experiences (also known as the replay approach). Generative replay, that is using generative models to provide replay patterns on demand, is particularly intriguing, however, it was shown to be effective mainly under simplified assumptions, such as simple scenarios and low-dimensional benchmarks. In this paper, we show that, while the generated data are usually not able to improve the classification accuracy for the old classes, they can be effective as negative examples (or antagonists) to learn the new classes, especially when the learning experiences are small and contain examples of just one or few classes. The proposed approach is validated on complex class-incremental and data-incremental continual learning scenarios (CORe50 and ImageNet-1000) composed of high-dimensional data and a large number of training experiences: a setup where existing generative replay approaches usually fail.

## 1 INTRODUCTION

The majority of neural networks training approaches assume that is feasible to build a set of independent and identically distributed (i.i.d.) samples to train the model. This assumption is in contrast with biological learning since intelligent beings observe the world as an ordered sequence of highly correlated data. When state-of-the-art deep neural networks are trained continually, and the whole data cannot be accessed at once, the model suffers from the catastrophic forgetting problem (McCloskey & Cohen, 1989), and the knowledge about old data (old experiences) tend to be overwritten by new examples.

Storing part of past data and replaying them interleaved with new data proved to be an effective approach to mitigate forgetting (see Hayes et al. (2021) for a comprehensive survey). However, in some applications, the storage overhead together with privacy issues make replay techniques unfeasible. Therefore, *generative replay* has been recently explored, where a generative model is trained to produce data from past experiences (see Lesort et al. (2018) and Shin et al. (2017)). Besides solving the replay memory issue, generative replay can theoretically be capable of generating more general and novel examples not included in past experiences, thus potentially overcoming replay methods. Unfortunately, generative replay introduces much complexity due to the need for an interleaved incremental training of both a classifier and a generator. Moreover, generative models are usually complex and unstable to train, especially in incremental scenarios. Several researchers have shown that generative replay fails in complex CL scenarios with high-dimensional data (see Aljundi et al. (2019); Lesort et al. (2018) and van de Ven et al. (2020)) mainly due to the inaccuracies in the data generation that progressively grows across the experiences if a single generator is incrementally updated (see related works in section 5 for more details). The photocopy example helps to understand why. Let us consider a high-quality photocopy machine: when a picture is initially copied the output looks very similar to the original, but if the process is repeated several times by using as input the output of the previous step, some artifacts will soon appear and, after many iterations, the result will be highly compromised. Hence, even if some state-of-the-art models have been proved to be effective in generating also high dimensional data (Huang et al. (2018) Karras et al. (2019)), the continual training of such generators remains a challenging problem.

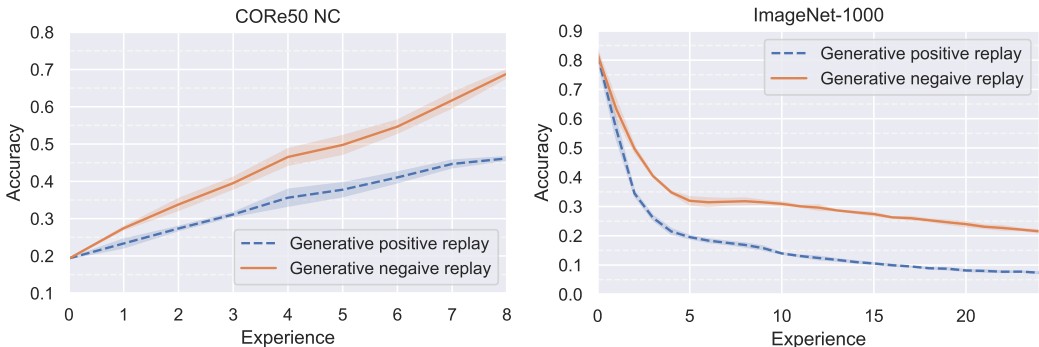

Figure 1: The proposed generative negative replay is compared with classical generative replay on two complex class incremental CL benchmarks (details in section 4.2). In both the benchmarks, using the same classifier, generator, and training procedure, negative replay performs significantly better.

Although generative models are a hot research topics and we can expect improved methods in the future, as of today we must deal with imperfect generated data and try to exploit them at best when a classifier is incrementally trained. The proposed approach, denoted as *Generative Negative Replay*, does not attempt to improve the knowledge of old classes using the generated data because it assumes that the data quality is not enough for this purpose. Nevertheless, it makes use of generated (latent) data as negative examples to better learn the classes of current experience, especially when the number of classes per experience is small and we incur in the "learning in isolation" problem.

We experimentally demonstrate, on complex benchmarks such as CORe50 and ImageNet-1000, where (positive) generative replay fails, that negative replay is effective to contrast the learning in isolation problem, allowing to train a classifier incrementally across a high number of experiences (see Figure 1). We also investigate the impact of data quality on negative replay with an ablation study (section 4.4) where negative examples are sampled from original past patterns (upper bound) and randomly generated.

## 2 PROBLEM FORMULATION

A continual learning (CL) problem consists of a number $N_E$ of experiences, each containing a subset of data that is only accessible during the corresponding experience:

$$\text{CL} = \{e_1, e_2, ..., e_{N_E}\}, \tag{1}$$

each of them is composed of several data points and the corresponding labels:

$$e_k = (\mathcal{X}_k, \mathcal{Y}_k), \quad \mathcal{X}_k = \{x_1^k, x_2^k, ..., x_{N_k}^k\}, \quad \mathcal{Y}_k = \{y_1^k, y_2^k, ..., y_{N_k}^k\} \tag{2}$$

where $x_i^k$ and $y_i^k$ are the data points and the associated labels contained in the $k$-th experience and $N_k$ is the number of samples in the $k$-th experience.

Let $\mathcal{D} = (\mathcal{X}, \mathcal{Y})$ be the entire dataset, then $\mathcal{X} = \bigcup_{i=1}^{N_E} \mathcal{X}_i$ and $\mathcal{Y} = \bigcup_{i=1}^{N_E} \mathcal{Y}_i$.

We can define three different scenarios for supervised continual learning (Maltoni & Lomonaco, 2019; van de Ven & Tolias, 2018): New Instances (NI), New Classes (NC), and New Instances and Classes (NIC). In NI (also known as domain incremental) all the classes are introduced in the first experience, and only new examples of the same classes are included in the following experiences. In NC (also known as class incremental) each experience contains only examples from classes never seen before. NIC is the combination of NI and NC, so each experience can be composed of new examples of already seen classes and/or examples from new classes. A formal definition of the above scenarios can be found in Appendix A.

Given the above definitions, our goal is to fit a function $f$, parametrized by $\Theta$, to the sequence of experiences. A naive approach is finding the best parameters $\Theta^*$ that minimizes:

$$\Theta^* = \arg\min_{\Theta} \mathcal{L}(f_\Theta(\mathcal{X}_i), \mathcal{Y}_i) \text{ for } i = \{1, ..., N_E\}, \tag{3}$$

where $\mathcal{L}(\cdot)$ is a loss function (e.g. cross entropy loss).

As first pointed out by McCloskey & Cohen (1989), this simple approach is prone to catastrophic forgetting, thus the model $f_\Theta$ is not able to learn the experiences $\{e_1, e_2, ..., e_{N_E}\}$ sequentially.

## 2.1 CONTINUAL LEARNING WITH GENERATIVE REPLAY

Generative replay requires to train simultaneously and incrementally a classifier and a generative model (Shin et al., 2017; Wu et al., 2018; Thandiackal et al., 2021) The generative model $g$, parametrized by $\Omega$ provides surrogate data similar to the past experiences' data. In the case of a conditional generative model (in which we can control the class of the generated data), the optimal parameters of the classifier can be derived using a replay memory as follows:

$$\Theta^* = \arg\min_{\Theta} \mathcal{L}(f_\Theta(\mathcal{X}_i \cup \mathcal{M}_i^x, \mathcal{Y}_i \cup \mathcal{M}_i^y) \text{ for } i = \{1, ..., N_E\}, \tag{4}$$

where $\mathcal{M}_i^x$ and $\mathcal{M}_i^y$ are the datapoints and labels contained in the replay memory during the training on experience $i$ when the replay memory is populated as:

$$\mathcal{M}_i^x \leftarrow g_\Omega(z_j|c_j); \;\; \mathcal{M}_i^y \leftarrow c_j; \;\; c_j \in \bigcup_{k=1}^{i-1} \mathcal{Y}_k, \;\; j = \{1, ..., R\}, \tag{5}$$

where $R$ is the number of generated replay patterns (size of memory), $z$ is a latent random input vector to the generative model, $c$ is a label sampled from the set of labels encountered in the past experiences, and "$\leftarrow$" indicates the insertion of an element in the memory.

The same generated data fed to the classifier can be used to control forgetting in the generative model as well. Instead of a generic generative model, suppose we have a conditional generative model composed of an encoder $q_\gamma$ parametrized by $\gamma$ and a decoder $p_\xi$ parametrized by $\xi$, such that $g_\Omega = p_\xi \circ q_\gamma$, $\Omega = (\gamma, \xi)$. The optimal parameters of the generative model can be obtained requiring that the generated data are similar (L2 loss) to the original ones:

$$\gamma^*, \xi^* = \arg\min_{\gamma, \xi} \|p_\xi(q_\gamma(\mathcal{X}_i \cup \mathcal{M}_i^x)|\mathcal{Y}_i \cup \mathcal{M}_i^y) - \mathcal{X}_i \cup \mathcal{M}_i^x\|_2^2 \;\; \text{for } i = \{1, ..., N_E\}, \tag{6}$$

where $q_\gamma(\mathcal{X}_i)$ is forced to follow a target distribution, typically $\mathcal{N}(0, 1)$.

## 3 GENERATIVE NEGATIVE REPLAY

As discussed before, generative replay is an appealing strategy for continual learning, but, to exploit it in complex scenarios with many experiences, we need to overcome the data degradation issue. Since this problem is not easily addressable on the generator side, we propose to circumvent it by changing the way the classifier makes use of generated data.

Let us suppose the classifier $f_\Theta$ can be divided into a feature extractor $f_\phi$, parametrized by $\phi$ and a classification head $c_\psi$ parametrized by $\psi$, so that $f_\Theta = c_\psi \circ f_\phi$, $\Theta = (\phi, \psi)$. The parameters $\psi$ of the classification head can be divided into $C$ groups, where $C$ is the number of classes. The groups, denoted as $(\psi^1, \psi^2, ..., \psi^C)$ represents the parameters associated to the connections between the features extracted by $f_\phi$ and the output neuron of the corresponding class.

For simplicity, let us assume that the feature extraction weights $\phi$ are frozen (after an initial pre-training) and, across the experiences, we only learn the classification head weights $\psi$. As explained in Section section 3.3, this assumption is not necessary and our experiments were carried out by learning both $\phi$ and $\psi$.

### 3.1 LEARNING CLASSES IN ISOLATION

Learning in isolation is one of the main causes of catastrophic forgetting, especially in the NC or NIC scenarios where only a limited number of classes are present in a single experience, and the parameters of the classification head are learned without negative examples that counteract the "greediness" of the optimization. As an example, let us consider an NC scenario where only one class is present in each experience. Suppose that $c$ is the only class in the experience $k$, then the

best way to optimize the model is to change the parameters $\psi^c$ to maximize the output of the output neuron $c$ for every input and change the rest of $\psi^j$, $j \neq c$ to minimize the output for remaining classes. This still holds if in the experience are present only a few classes, since the model is only optimized to discriminate between the present classes and has no interest in maintaining the past acquired knowledge.

### 3.2 POSITIVE AND NEGATIVE REPLAY

Replay can be used to counteract the learning in isolation problem, however, when the replay data comes from a generative model, the data quality degradation has a negative impact on the classifier training. The aforementioned problem is typical of the standard generative replay approach (hereafter denoted as *generative positive replay*), where replay data is used by the classifier in the same manner of the current experience's data, and therefore the classification head's weights associated to the replay classes are optimized based on the replay data.

On the contrary, in the proposed *generative negative replay* approach the replay patterns are used to counteract the detrimental effects of the training in isolation but they are not used to modify the parameters $\psi$ associated with the replay classes. The key idea (validated experimentally) is that the generated patterns are valid antagonists to mitigate the learning in isolation problem, but their quality is not enough to improve the knowledge of classes originally learned on real data. It is well known that one class learning approaches are in general less effective than discriminative learning because the presence of both positive and negative examples allows to better characterize the classification boundaries (Hempstalk & Frank, 2008). Therefore, the proposed approach exploits generated data to constrain the classification boundary and to avoid that the real data in the current experience pull it too much in their direction.

### 3.3 TRAINING A CLASSIFIER WITH GENERATIVE NEGATIVE REPLAY

The idea of generative negative replay is quite general can be used in conjunction with different continual learning classification approaches and scenarios (NI, NC, NIC). To avoid replay data (i.e. negative examples) alter the knowledge of the already learned classes, the gradient accumulation can be selectively blocked during the backward pass. The general idea is illustrated in Figure 2. While the original examples ($\mathcal{X}_i$) normally flows forward and backward throughout the model, the replay examples ($\mathcal{M}_i^x$) are passed forward, but, before the backward pass, the loss tensor is masked at the class level by resetting the gradient components corresponding to the classes in $\mathcal{M}_i^y$. The negative replay implementation illustrated in Figure 2 is discussed in more details in Appendix C where preliminary tests on Core50 NC are also included.

Hereafter, we provide an alternative implementation embedded in AR1 algorithm (Maltoni & Lomonaco, 2019), whose update mechanism for the classification head weights allows very simple and efficient integration of negative replay. AR1 is a flexible continual learning approach that can achieve state-of-the-art accuracy on complex CL benchmarks. In Appendix E, AR1 is shown to outperform several recent CL algorithms of the difficult ImageNet-1000 benchmark proposed by Masana et al. (2020).

AR1 uses different mechanisms to learn the classification head and the feature extractor weights. The feature extraction weights $\phi$ are protected against forgetting: i) through the Synaptic Intelligence regularization technique (Zenke et al., 2017) or ii) using a replay memory with a small learning rate (denoted as AR1free in Pellegrini et al. (2020)). The classification head weights $\psi$ are managed by CWR. CWR is a simple method aimed at addressing the score bias problem produced by imbalance learning during continual learning (Belouadah et al., 2020). CWR (Maltoni & Lomonaco, 2019) maintains a copy of the weights of the classification head of the previous experience ($\psi'$) and at the start of each experience the classification head is reset and only weights of classes of the current experience are loaded from $\psi'$. At the end of the experience, a weight consolidation phase takes place, where the weights $\psi$ learned in the current experience are consolidated with the weights $\psi'$. This is the point where positive and negative replay behaves differently.

In particular, during the consolidation phase, for each parameter group $\psi^c$ associated to a class $c$ belonging to the current experience ($c \in \mathcal{Y}_k \cup \mathcal{M}_k^y$), the mean of all the parameter group $\mu(\psi^c)$ is calculated, and subtracted to all the parameters in the group, in order to force zero mean: $\psi^c =$

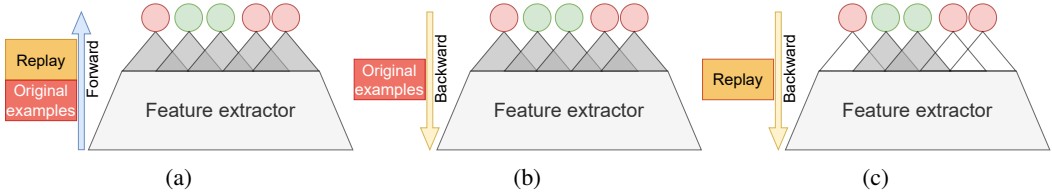

Figure 2: Graphical representation of the negative replay idea. Green output neurons represent the classes present in the current experience, while red output neurons represent the replay classes. During forward (a) both the replay data and the original data from the current experience flow through the network. During backward, the original data flow through all the neurons of the classification head (b), while the replay data contribution is masked and data only flows through the neurons of classes belonging to the current experience (c).

$\psi^c - \mu(\psi^c)$. This prevents class bias problems due to the different magnitudes of the weights. Then, there are three possibilities, based on $c$:

1. $c$ is a new class never seen before ($c \in \mathcal{Y}_k \land c \notin \bigcup_{i=1}^{k-1} \mathcal{Y}_i$): in this case $\psi^c$ is maintained as is.

2. $c$ is a class seen before ($c \in \mathcal{Y}_k \land c \in \bigcup_{i=1}^{k-1} \mathcal{Y}_i$): the consolidation step is applied, so $\psi^c = \frac{\psi'^c \cdot w_{past_c} + \psi^c}{w_{past_c} + 1}$ where $w_{past_c}$ is a parameter that balances the contribution of the past w.r.t. the present, calculated as follows:

$$w_{past_c} = \sqrt{\frac{past_c}{current_c}}, \tag{7}$$

   where $past_c$ is the number of data points of class $c$ encountered in past experiences, while $current_c$ is the number of data points of class $c$ present in the current experience.

3. $c$ is not in the current experience but is a replay example ($c \notin \mathcal{Y}_k \land c \in \mathcal{M}_k^y$):
   - in case of positive replay apply consolidation (step 2).
   - in case of negative replay $\psi^c$ is substituted with $\psi'^c$ (no contribution to the parameters $\psi^c$ from replay examples).

The pseudo-code of the above weigh consolidation algorithms is reported in Appendix B. It is worth noting, that in the proposed embedding of negative replay in AR1, the replay pattern can alter the feature extraction weights since CWR weight consolidation only "protects" the classification head However, in our experiments, we found that a more complex embedding of negative replay in AR1 where we block the gradient propagation for negative patterns throughout the feature extraction layers performs very similarly, and therefore we opted for simplicity.

## 4 EXPERIMENTS AND RESULTS

In this section, we describe the experimental setup used to validate the proposed negative replay. We focus on difficult continual learning scenarios, where data is high-dimensional, non-i.i.d. and the number of experiences is very large. Negative replay is compared with alternative strategies (e.g. positive replay) and the role of quality of generated data is investigated by also using, as negative replay patterns, real and random data.

### 4.1 EXPERIMENTAL SETUP

**Datasets** We performed our experiments on the CORe50 dataset (Lomonaco & Maltoni, 2017) and ImageNet-1000 dataset (Deng et al., 2009). CORe50 dataset was specifically collected for continual learning (NI, NC, and NIC scenarios) and is composed of small video sessions (about 300 frames) of 50 objects taken from the point of view of a person that handles them in the hand. Every class has 11 video sessions (a total of about 3,300 images) with different backgrounds and illuminations. Eight video sessions for each class are used for training, and 3 for testing. Images have size 128×128

pixels. ImageNet is composed of 1,000 classes with about 1,000 patterns per class for training and 100,000 images for testing. All images are resized to 224×224 pixels.

**Classifier architecture**    AR1 algorithm was used with Synaptic Intelligence (SI) regularization when trained without replay, and without protection on the feature extraction weights (AR1free) in case of positive and negative replay. In the experiments with CORe50 dataset we follow Maltoni & Lomonaco (2019) and Lomonaco et al. (2020) by employing a MobileNetV1 network (Howard et al., 2017). As in Pellegrini et al. (2020) and van de Ven et al. (2020), we opted for latent replay, that is replaying latent activations instead of input data. As described in Pellegrini et al. (2020), the choice of the latent replay layer is related to a tradeoff between accuracy and efficiency. For CORe50 experiments, as in Pellegrini et al. (2020), we used the `conv5_4` layer as latent replay layer, and the classifier was pretrained on ImageNet-1000. We also substituted all the batch normalization layers of the network with batch renormalization (Ioffe, 2017). For ImageNet-1000 we use a ResNet-18 (He et al., 2016) architecture. Following the benchmark proposed by Masana et al. (2020) the model was not pretrained. To maintain compatibility with the experiments on CORe50, even on ImageNet-1000 we use latent replay, setting the replay layer on the fourth residual block of the network (after `conv4_x` using He et al. (2016) nomenclature).

**Generative model architecture**    For the choice of a generative model we initially focused on two state-of-the-art approaches whose implementations are open source (van de Ven et al., 2020; Shin et al., 2017; Ayub & Wagner, 2021). However, since they were designed to work in simpler settings (with a lower data dimensionality and a smaller number of experiences), we were not able to port and scale them to our complex setups. Therefore, we implemented a generative model by trying to combine the most promising techniques and ideas from different sources and control its overall memory/computation complexity. In particular, taking inspiration from van de Ven et al. (2020) we use a Variational Autoencoder (VAE) Kingma & Welling (2014) model, but unlike van de Ven et al. (2020) we opted for a conditional VAE (cVAE) configuration Sohn et al. (2015). Moreover, we partially blend the generator (encoder) with the classifier model: both the networks share the same feature extractor $f_\phi$. Finally, instead of generating raw data, we generate activations at an intermediate "latent" level as suggested by van de Ven et al. (2020). A detailed discussion on the architecture of the generator is provided in Appendix D, including a pseudo-code that highlights the details of the interleaved training of the generator and the classifier.

## 4.2 EXPERIMENTS ON THE NC SCENARIO

The first round of experiments has been performed on the NC scenario using CORe50 and ImageNet-1000. For CORe50 the benchmark is composed of 9 experiences: the first one contains 10 classes while the following contains five classes each. We used a replay memory of 1,500 patterns, and (for generative replay) we inserted in each minibatch, of size 128, 14 replay patterns, and 114 patterns from the current experience. We train both the classifier and the generator for 4 epochs for each experience. Hyper-parameters of the classifier and generator are reported in Appendix F and Appendix G respectively.

For ImageNet-1000 the benchmark follows the one proposed by Masana et al. (2020): the dataset is divided into 25 experiences of 40 classes each. We used a replay memory of 20,000 patterns, and (for generative replay) we inserted in each minibatch, of size 128, 36 replay patterns, and 92 patterns from the current experience. We did not expect negative replay to perform well in this setup, because each experience already contains 40 classes and, therefore, the learning-in-isolation problem is here marginal. Nevertheless, we were interested in understanding if, in this setup, negative replay hurts the learning process or still provides minor benefits.

The results are shown in Figure 3 and Table 1. In CORe50 the baseline with no replay (using the AR1 algorithm) reaches a final accuracy of about 60% while using replay raises the accuracy to more than 70% (Positive Replay Original Data - PR-OD). These were expected to be the lower and upper bounds of this experiment, respectively. However, because of the data degradation problem, performing positive replay with generated data (Positive Replay Generated Data - PR-GD) performed significantly worse than the case with no replay. Using replay in a negative manner with generated data, as proposed in this work (NR-GD), only slightly decreases the final accuracy with respect to the upper bound PR-OD.

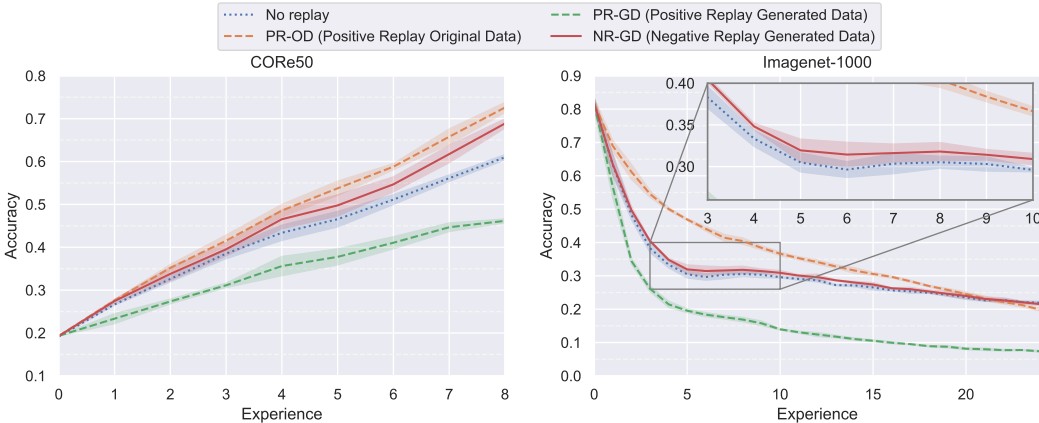

Figure 3: Overall accuracy on CORe50 NC scenario, using the whole test set (even at intermediate experiences) as defined in the CORe50 protocol (Lomonaco & Maltoni, 2017) (left), and on ImageNet-1000 using a growing test set as defined by Masana et al. (2020) (right). For a direct comparison of the two benchmarks, a plot of the experiments on CORe50 NC using a growing test set is included in Appendix H. Every experiment is averaged over 3 runs using different seeds and class order. The standard deviation is reported in light colors. Better viewed on a computer monitor.

| Method | CORe50 | ImageNet-1000 |
|---|---|---|
| No Replay | $41.68 \pm 0.62$ | $31.91 \pm 0.17$ |
| PR-OD (upper bound) | $47.02 \pm 0.45$ | $38.02 \pm 0.08$ |
| PR-GD | $34.05 \pm 0.29$ | $18.29 \pm 0.07$ |
| **NR-GD** | $\mathbf{44.63 \pm 0.77}$ | $\mathbf{32.74 \pm 0.17}$ |

Table 1: Average accuracy on all the experiences for the CORe50 and ImageNet-1000 NC scenarios.

For ImageNet-1000, due to the complexity of the experiment and the fact that the network is fully trained only during the first experience (blocked after `conv4_x` in the following experiences) the final accuracy are quite similar for all the methods (except PR-GD that performed far worse). However, in the first 10 experiences some differences can be appreciated: see the insect view in Figure 3-right. The impact of the generated data quality on negative replay is more evident in Table 1: using negative replay with generated data (in this case highly degraded) improve the average accuracy (calculated as the mean of the accuracy after each experience) of more than 24 points and the final accuracy of more than 10 points w.r.t. using replay data in a positive manner. Furthermore, even if in this scenario the advantage of negative generative replay is little with respect to the no replay case, we note that negative replay is not hurting the training process even in scenarios where learning in isolation is not an issue.

### 4.3 EXPERIMENTS ON THE NIC BENCHMARK

CORe50 NIC-391 protocol is composed of 391 learning experiences, each containing examples of a single class (300 frames of a short video). This scenario is particularly challenging and prone to learn-in isolation issues, hence we may expect the role of replay to be more important here. In this scenario, we used a replay memory of only 300 patterns. The minibatch size is 128, and when generative replay is employed, we generate 64 patterns for every mini-batch (plus 64 from the current experience). Hyper-parameters of the classifier and generator are reported in Appendix F and Appendix G respectively.

The results are shown in Figure 4 and they are quite in line with the previous experiment, but here the accuracy gaps grow and the benefit of replay is more evident. The proposed negative replay with generated data (NR-GD) performs quite well, about 10 points better than with no replay and just less than 5 points worse than positive replay with real data, the upper bound. Using generated data in a positive manner (PR-GD) is here even worse than in the NC case, because the data degradation

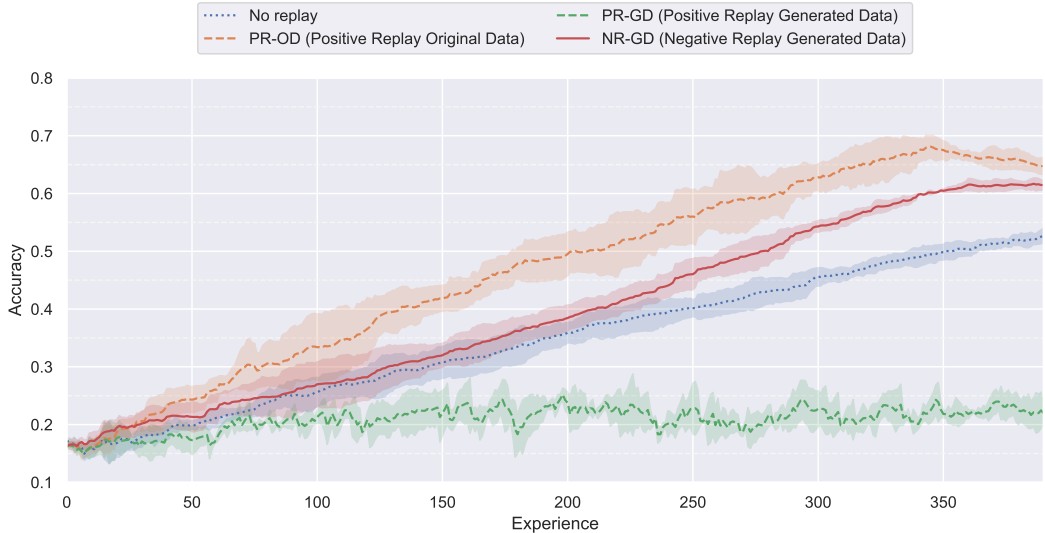

Figure 4: Overall accuracy on CORe50 NIC391 scenario, using the whole test set as defined in the CORe50 protocol (Lomonaco & Maltoni, 2017). Every experiment is averaged over 3 runs using different seeds and class order. The standard deviation is reported in light colors. better viewed on a computer monitor.

is amplified with so many iterations: PR-GD is losing 30 points w.r.t. not using replay at all, and performs about 40 points worse than using the same replay data with the proposed generative negative replay approach.

## 4.4 ABLATION STUDY

The effect of generated data quality on negative replay is investigated by performing two further experiments: NR-OD uses original data (max. quality) for negative replay, while NR-RD uses randomly generated replay data, obtained by uniform random sampling in the latent replay layer and assigning to each data point a random class label. Since in our experiments we replay hidden features, in order to produce reasonable replay data we first calculated the range of latent activations on a sample dataset, and then we set our random generator to produce values in the range: 0 (since we use ReLU activation functions) - $90th$ percentile of the real activation values. We used CORe50 NC and CORe50 NIC in these experiments.

The results are reported in Table 2. Surprisingly, even with random replay data (that we assume to be the worst degradation possible), negative replay is still able to perform better than no replay. Furthermore, the difference between original and generated data is minimal, thus proving that negative replay is tolerant in terms of data quality. Note that in both the experiment using random data with negative replay performs way better than using generated data in a classical (positive) manner (PR-GD in previous figures). Comparisons in all the benchmarks of all the experiments (positive and negative replay with original, generated, and random data) are reported in Appendix H.

| Method | CORe50 NC | CORe50 NIC |
|--------|-----------|------------|
| No Replay | $60.99 \pm 0.49$ | $52.71 \pm 1.02$ |
| NR-OG | $68.60 \pm 1.38$ | $67.93 \pm 0.31$ |
| NR-GD | $68.87 \pm 0.88$ | $61.46 \pm 0.67$ |
| NR-RD | $64.05 \pm 0.71$ | $58.85 \pm 0.58$ |

Table 2: Final accuracy on CORe50 NC and NIC using original (NR-OD), generated (NR-GD), and random (NR-RD) data with negative replay. The results with no replay are reported as references. Every experiment is averaged over 3 runs using different seeds and class order.

## 5 RELATED WORKS

The use of negative examples to learn more discriminative class boundaries can be traced back to *one-class support vector machines (SVM)* (Chen et al., 2001), where the data points belonging to the other classes in the training set are used as negative examples. Malisiewicz et al. (2011) proposed using an ensemble of one-class SVMs instead of a single multi-class classifier. This approach operates in a scenario which is similar to the experiments on the CORe50 NIC benchmark, whose experiences contains only one class and all the replay data points (possibly belonging to many past encountered classes) are used as negative examples. The use of negative examples can also be seen as a kind of *contrastive learning* (Khosla et al., 2020), where negative examples are used to cluster embeddings of data points of the same class while moving away embeddings of data from different classes.

Masking parts of a neural network has been experimented before in continual learning. Wortsman et al. (2020) masked the weights of a randomly initialized neural network in order to find a sub-network that yields good performance for a particular task. The loss masking proposed for standalone negative replay introduced in section 3.3 and Appendix C (without using any continual learning strategy) is similar to the masking method proposed by Masana et al. (2021). In that work, each feature can be used normally, masked (not used), or used only during forward (no modification of the related parameters during network update).

Generative replay for continual learning was first introduced by Shin et al. (2017) who proposed Deep Generative Replay (DGR), using a generative adversarial network (GAN) (Goodfellow et al., 2014). Many works on generative replay (Wu et al., 2018; Ostapenko et al., 2019) use GANs as generative models, but GANs are usually slow and complex to train, even in non-incremental scenarios. Ayub & Wagner (2020) proposed to use autoencoders, however that approach requires maintaining a generative model for every experience, making it not scalable to long incremental sequences. Kemker & Kanan (2018) and van de Ven et al. (2020) proposed continual learning framework inspired to biological brain functionalities and memories. In particular, (van de Ven et al., 2020) showed significant results in continual learning scenarios with dozens of experiences. However, this approach was not tested on high-dimensional data and in much bigger scenarios.

## 6 CONCLUSION

In this paper, we addressed the problem of continual learning with generative replay, focusing on the obstacles of generative replay in complex scenarios. Our experience confirms that incrementally training a generator over a long number of experiences with high dimensional data is a very challenging problem and remains an open issue. Therefore, instead of trying to design a better generative model, we focused on classifier training. We found that even inaccurate replay data can be useful to contrast the learning in isolation problem, especially in scenarios where only a limited number of classes is present in each experience. We called this approach negative replay since the replay data is used as negative examples when the model is trained with data from the current experience. We validated negative replay using complex continual learning scenarios, with high dimensional data and hundreds of incremental experiences. The results show that using negative replay largely improves classification performances w.r.t. using the generated data in a traditional fashion. We also investigated the impact of generated data quality, by considering the two extremes of using original data and random data for negative replay, and, surprisingly, we found that negative replay is effective even using random replay data.

Since negative replay can be easily applied to other continual learning strategies (besides AR1), we believe that many other CL approaches may benefit from our proposal, especially when complex scenarios are employed. Moreover, negative replay could be used in the pre-training phase of large models, possibly making them more robust to noise or degraded data. Finally, is worth noting that dealing with imprecise replay data can be viewed as a biological feature since human's memory is far from being accurate, but is thought to be essential to consolidate learning (van de Ven et al., 2020), therefore investigating the role of negative replay-like mechanisms in biological learning could be an interesting research direction for computer scientists and neuroscientists.

REPRODUCIBILITY STATEMENT

The source code of the project, alongside configuration files for the reproducibility of experiments, is included in the additional materials. A public version of the code, based on the Avalanche framework (Lomonaco et al., 2021) will be released upon publication. The datasets used are publicly available and can be downloaded from the respective official websites[1,2]. Both the datasets are preprocessed normalizing them using statistics derived from ImageNet-1000. Images from ImageNet-1000 were randomly cropped and resized to the final dimension of $224 \times 224$, then horizontally flipped with a probability of $50\%$.

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

## A    NI, NC, AND NIC DEFINITIONS

Given the definitions of section 2, the NI (New Instances), NC (New Classes), and NIC (New Instances and Classes) continual learning scenarios can be defined based on the labels $\mathcal{Y}_k$ contained in the experiences ($k \in \{1, ..., N_E\}$ with $N_E$ the total number of experiences) as follows:

**New instances (NI)** also known as domain-incremental learning, where all the labels are known from the first experience, and the successive experiences, only new instances of the same classes are included. Formally, we could define the NI scenario as:

$$\mathcal{Y}_1 \cap \mathcal{Y}_k = \mathcal{Y}_k \quad \text{for } k = \{1, ..., N_E\}, \tag{8}$$

meaning that every possible label of the entire dataset must be present in the first experience.

**New classes (NC)** also known as class-incremental learning, where each experience includes data of classes not present in any other experience. Formally, we can define the NC scenario as:

$$\mathcal{Y}_k \cap \bigcup_{i=1}^{k-1} \mathcal{Y}_i = \emptyset \quad \text{for } k = \{2, ..., N_E\}. \tag{9}$$

**New instances and classes (NIC)** where a new experience can contain already seen classes, new classes, or a mix of the two. This is the most natural scenario since in the real world an agent may sense both known and unknown objects. Formally the NIC scenario can be defined as:

$$\exists k : \mathcal{Y}_k \cap \bigcup_{i=1}^{k-1} \mathcal{Y}_i \neq \emptyset \text{ and } \exists j : \mathcal{Y}_1 \cap \mathcal{Y}_j \neq \mathcal{Y}_j. \tag{10}$$

Meaning that there is at least one experience that contains classes already seen in the past (left part) and at least one experience that contains classes not present in the first experience (right part).

## B    PSEUDO-CODE OF THE WEIGHT CONSOLIDATION PHASE

---
**Algorithm 1** Weight consolidation
---
**Require:** $\psi$, $\psi'$, $\mathcal{Y}_e$, $\mathcal{M}_e^y$
1: **for** each class $c \in \mathcal{Y}_e \cup \mathcal{M}_e^y$ **do**
2:      $\psi^c = \psi^c - \mu(\psi^c)$
3:      **if** $c \in \mathcal{Y}_e \wedge c \in \bigcup_{i=1}^{e-1} \mathcal{Y}_i$ **then**
4:          $\psi^c = \frac{\psi'^c \cdot w_{past_c} + \psi^c}{w_{past_c} + 1}$
5:      **end if**
6:      **if** $c \notin \mathcal{Y}_e \wedge c \in \mathcal{M}_e^y$ **then**
7:          **if** positive replay **then**
8:              $\psi^c = \frac{\psi'^c \cdot w_{past_c} + \psi^c}{w_{past_c} + 1}$
9:          **end if**
10:        **if** negative replay **then**
11:            $\psi^c = \psi'^c$
12:        **end if**
13:      **end if**
14: **end for**
15: $\psi' = \psi$
---

## C    STANDALONE NEGATIVE REPLAY IMPLEMENTATION

To validate the proposed negative replay approach, we implemented a version of negative replay that does not depend on any specific continual learning strategy. As discussed in the main text, the main idea behind negative replay is to avoid replay data altering the knowledge of the already learned classes. In other words, the replay data cannot change the weights of the already learned classes.

This behavior can be obtained by selectively blocking the gradient accumulation during the backward pass. The original examples ($\mathcal{X}_i$) flow normally through the network, accumulating the gradient in all the feature extractor weights ($\psi$). On the other hand, the replay data points ($\mathcal{M}_i^x$) participate in the loss calculation, but their backward contribution is limited to the weights associated with the output neurons of the original data classes. The loss tensor associated to replay data is thus masked at class level, resetting the gradient contribution to zero for all the classes in $\mathcal{M}_i^y$. The procedure is graphically explained in Figure 5.

Note that the proposed standalone negative replay implementation does not provide any specific mechanism to counteract catastrophic forgetting problem, and since it applies negative replay on top of the naive CL strategy we denote the resulting strategy as *naive negative replay*.

In line with the experiments included in the main text, we compared naive negative replay (NR-GD) against naive with no replay (lower bound), naive with positive generative replay (PR-GD), and naive with positive replay with original data (PR-OD, upper bound). We used the CORe50 NC benchmark to perform this experiment.

The results are shown in Figure 6. As expected, the overall accuracy with naive is lower than with AR1, but the relative ranking of the different replay approaches in maintained. In particular, negative replay performs better w.r.t. no replay (naive) or using the generated data in a positive manner (naive PR-GD). In this scenario the gap between negative replay with generated data and positive replay with real data is larger than when using AR1, because replay is the only methods to contrast forgetting and the replay data quality is more relevant.

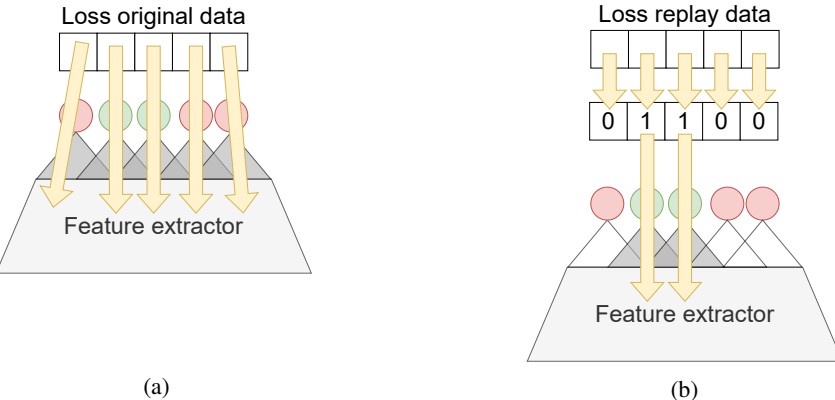

Figure 5: Graphical representation of the naive negative replay. The output neurons are depicted in green for neurons associated with original data, and in red for output neurons associated to replay data. The loss vector of original data is backpropagated through all the weights of the network (yellow arrows), so the gradient is accumulated by every weight. On the other hand, the loss vector of replay data is masked, and the loss is only backpropagated through the output neurons corresponding to the original classes. Thus, the gradient is accumulated only on the classification head weights associated with the classes present in the current experience.

## D    DETAILS OF THE GENERATIVE MODEL IMPLEMENTATION

We designed our generative model using different insights from previous works in the fields, bringing together different ideas and proposals. We extensively tested the generative model alone to find the better combination of building blocks that yield the best performance. Our design choices have been also influenced by the computation complexity since our aim is to develop a (near) real-time system.

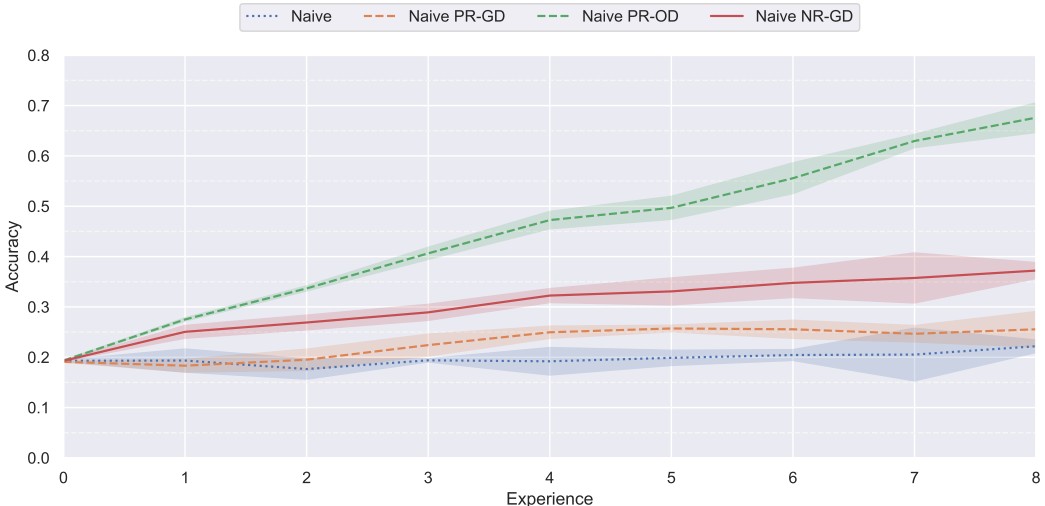

Figure 6: Overall accuracy on CORe50 NC scenario of the naive negative replay, using the whole test set as defined in the CORe50 protocol Lomonaco & Maltoni (2017). Every experiment is averaged over 3 runs using different seeds and class order. The standard deviation is reported in light colors. Better viewed on a computer monitor.

This is a particularly hard constraint since many incremental generative replay methods are based on generative adversarial networks (GANs) (Goodfellow et al., 2014), which notably have long training phases and often suffer from instabilities due to the adversarial nature of the training procedure.

As discussed in the main text, we took inspiration from some state-of-the-art methods, trying to combine promising techniques and ideas from different sources. Taking inspiration from van de Ven et al. (2020) we use a Variational Autoencoder (VAE) Kingma & Welling (2014) model, but unlike van de Ven et al. (2020) we opted for a conditional VAE (cVAE) configuration Sohn et al. (2015). So, while in van de Ven et al. (2020) a mixture of Gaussian is used to sample latent vectors and soft labels are provided to the classifier itself, in our approach the latent vector is sampled from the normal distribution $\mathcal{N}(0,1)$ and conditioned to the desired class. This results in a faster and less complicated sampling of a replay pattern. Moreover, as in van de Ven et al. (2020) we partially blend the encoder part of the generative model with the classifier model: in fact both the network share the same feature extractor $f_\phi$. For the classifier, this branch is connected with the classification head $c_\psi$, while, for the generator, it is connected with some other layers that transform the feature into a latent vector $z$. The bifurcation is located in the latent replay layer. The resulting on-the-loop training of the generative model is consistent with brain structures and neuroscience's findings (van de Ven et al., 2020).

Since we use a cVAE, the objective for the generative model can be expressed as:

$$\gamma^*, \xi^* = \arg\min_{\gamma,\xi}[-\mathbb{E}_{z \sim q_\gamma(z|x_i^k)}[\log p_\xi(z|y_i^k)] + D_{KL}(q_\gamma(z|x_i^k)||p(z))], \tag{11}$$

where $(x_i^k, y_i^k)$ are the data point and the label of the $i$-th pattern of the $k$-th experience, and the $D_{\mathrm{KL}}$ term represents the Kullback-Leibler divergence between the latent space distribution and the target distribution $p(z) = \mathcal{N}(0,1)$.

The two terms of Equation 11 determine two losses:

$$\mathcal{L}_{recon} = \|x_i^k - p_\xi(q_\gamma(x_i^k))\|_2^2 \tag{12}$$

$$\mathcal{L}_{KL} = D_{\mathrm{KL}}(q_\gamma(x_i^k)||\mathcal{N}(0,1)) \tag{13}$$

We also add another loss term, denoted as *classification loss*, which is similar to the classifier loss adopted in the AC-GAN model (Odena et al., 2017). The rationale is to guide the generative model to produce data that are not only visually similar to the original ones (L2 loss) but that is also classified

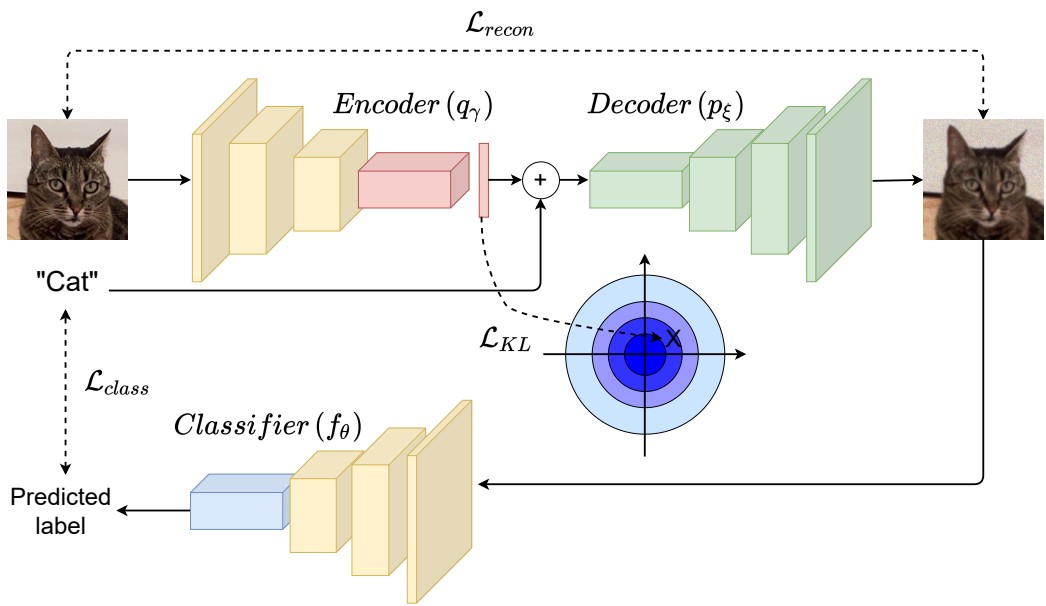

Figure 7: A visual schema of the generative model training. Losses are represented by dashed arrows. The shared branch of the classifier and the encoder are depicted using the same color (yellow). The encoder and the classifier's additional layers are drawn in red and blue respectively.

by the current classifier in the same way. Hence, we use $f_\Theta$ as "auxiliary" classifier, adding the following term to the generator's loss:

$$\mathcal{L}_{class} = -\log f_\Theta(y_i^k | p_\xi(q_\gamma(x_i^k) | y_i^k)), \qquad (14)$$

which represents a typical negative log-likelihood classification loss. Note that the parameters $\Theta$ of the classifier are not trained in this phase, since only the generative model is updated. Overall, the generative model is trained using the following loss function:

$$\mathcal{L}_{GM} = \mathcal{L}_{recon} + \beta \mathcal{L}_{KL} + \eta \mathcal{L}_{class}, \qquad (15)$$

where $\beta$ is a hyper-parameter inspired to the $\beta$-VAE framework (Higgins et al., 2017), and $\eta$ is a hyper-parameters that weights the importance of the classification loss.

A visual representation of generative model training is shown in Figure 7.

To keep notation light, in the equations above the replay memory is not used, but it is trivial to include patterns from the replay memory, since there is no distinction in the generative model training procedure between current and replay data.

Note that the utilization of raw images is not mandatory for the method, and any intermediate (or latent representation) can be used, making our proposal compatible with latent replay methods (Pellegrini et al., 2020; van de Ven et al., 2020). In fact, in the case of latent replay, the data points $x_i^k$ in the above equations can be simply substituted with $f_{\phi'}(x_i^k)$, where $f_{\phi'}$ is the set of feature extraction layers before the latent replay layer.

The blending of a part of the generative model into the classifier poses some difficulties in the training, especially regarding the balancing of the two models and how to train each of them without destructive inference on the other. After some initial experiments, we opted for blocking model parameters when the other model is trained. Detailed pseudo-code for the proposed negative generative replay strategy is provided in Algorithm 2.

---

**Algorithm 2** Generative negative replay

---

1:  $f_\Theta \leftarrow$ RANDINIT or PRETRAINED
2:  $g_\Omega \leftarrow$ RANDINIT or PRETRAINED
3:  $\mathcal{M}^x \leftarrow \emptyset, \;\; \mathcal{M}^y \leftarrow \emptyset$
4:  $R =$ memory size
5:  **for** each $k$ from 1 tp $N_E$ **do**
6:     **if** $k > 1$ **then**
7:         SAMPLE $\{z_1, ..., z_R\} \sim \mathcal{N}(0,1)$
8:         SAMPLE $\{c_1, ..., c_R\} \sim \bigcup_{t=1}^{k-1} \mathcal{Y}_t$
9:         BLOCK   generator parameters $(\gamma, \xi)$
10:        POPULATE $\mathcal{M}_k^x = p_\xi(z_j|c_j)), \; j = \{1, ..., R\}$
11:        POPULATE $\mathcal{M}_k^y = \{c_1, ..., c_R\}$
12:     **end if**
13:     `# classifier training`
14:     $\psi' = \psi$
15:     BLOCK generator parameters $(\gamma, \xi)$
16:     UNLOCK classifier parameters $(\phi, \psi)$
17:     $\phi^*, \psi^* =$ OPTIMIZE$(f_\Theta, \mathcal{X}_k \cup \mathcal{M}_k^x, \mathcal{Y}_k \cup \mathcal{M}_k^y)$ using Equation 4
18:     WEIGHTCONSOLIDATION$(\psi, \psi', \mathcal{Y}_k, \mathcal{M}_k^y)$ (see Algorithm 1)
19:     `# generator training`
20:     BLOCK  classifier parameters $(\phi, \psi)$
21:     UNLOCK  generator parameters $(\gamma, \xi)$
22:     $\gamma^*, \xi^* =$ OPTIMIZE$(g_\Omega, \mathcal{X}_k \cup \mathcal{M}_k^x, \mathcal{Y}_k \cup \mathcal{M}_k^y)$ using Equation 15
23: **end for**

---

# E   VALIDATION OF AR1 ON IMAGENET-1000

To validate the chosen AR1 algorithm we performed a test on a competitive benchmark on ImageNet-1000, following the NC benchmark proposed by Masana et al. (2020), which is composed of 25 experiences, each of them containing 40 classes. The benchmark is particularly challenging due to a large number of classes (1,000), the incremental nature of the task (with 25 experiences), and the data dimensionality of $224 \times 224$ (as with ImageNet protocol).

With this experiment we want to assess the performance of AR1 in a complex continual learning scenario, validating the choice of AR1 as the baseline algorithm on which the tests on negative replay are conducted. In this experiment, we tested AR1 against both regularization-based methods (Dhar et al., 2019; Kirkpatrick et al., 2017; Li & Hoiem, 2016) and replay-based approaches (Belouadah & Popescu, 2019; Castro et al., 2018; Chaudhry et al., 2018; Hou et al., 2019; Rebuffi et al., 2017; Wu et al., 2019). We use the same classifier (ResNet-18 (He et al., 2016)) and the same memory size for all the tested methods (20,000 patterns, 20 per class); for the regularization-based approaches, the replay is added as an additional mechanism.

For AR1, we trained the model with an SGD optimizer. For the first experience, we used an aggressive learning rate of 0.1 with momentum 0.9 and weight decay of $10^{-4}$. We multiply the initial learning rate by 0.1 every 15 epochs. We trained the model for a total of 45 epochs, using a batch size of 128. For all the subsequent experiences we used SGD with a learning rate of $5 \cdot 10^{-3}$ for the feature extractor's parameters $\phi$ and $5 \cdot 10^{-2}$ for the classifier's parameters $\psi$. We trained the model for 32 epochs for each experience, employing a learning rate scheduler that decreases the learning rate as the number of experiences progresses. This was done to protect old knowledge against new knowledge when the former is more abundant than the latter. As in the first experience, the batch size was set to 128, composed of 92 patterns from the current experience and 36 randomly sampled (without replacement) from the replay memory.

The results are shown in Table 3. Replay-based methods exhibit the best performance, with iCaRL and BiC exceeding a final accuracy of 30%. AR1 outperforms all the baselines (33.1%), demonstrating the validity of this approach also in difficult continual learning benchmarks. However, considering

| Method | Final Accuracy |
|---|---|
| Fine Tuning (Naive) | 27.4 |
| EWC-E (Kirkpatrick et al., 2017) | 28.4 |
| RWalk (Chaudhry et al., 2018) | 24.9 |
| LwM (Dhar et al., 2019) | 17.7 |
| LwF (Li & Hoiem, 2016) | 19.8 |
| iCaRL (Rebuffi et al., 2017) | 30.2 |
| EEIL (Castro et al., 2018) | 25.1 |
| LUCIR (Hou et al., 2019) | 20.1 |
| IL2M (Belouadah & Popescu, 2019) | 29.7 |
| BiC (Wu et al., 2019) | 32.4 |
| **AR1** (Maltoni & Lomonaco, 2019) | **33.1** |

Table 3: Final accuracy on ImageNet-1000 following the benchmark of Masana et al. (2020) with 25 experiences composed of 40 classes each. For each method, a replay memory of 20,000 patterns is used (20 per class at the end of training). Results for other methods reported from Masana et al. (2020).

that top-1 ImageNet accuracy for a ResNet-18 when trained on the entire dataset is 69.76%[3], even for the best methods the accuracy gap in the continual learning setup is very large. This suggests that continual learning, especially in complex scenarios with a large number of classes and high dimensional data, is far to be solved, and further research should be devoted to this field.

---

[3]Accuracy taken from the torchvision official page: `https://pytorch.org/vision/stable/models.html`

# F   CLASSIFIER HYPER-PARAMETERS

## F.1   CORE50 NC

| | Hyper-parameter | Value |
|---|---|---|
| | optimizer | SGD |
| | momentum | 0.9 |
| | weight decay | $10^{-4}$ |
| Common | minibatch size | 128 |
| | SI (Synaptic Intelligence) $\lambda$ | $8 \cdot 10^5$ |
| | SI Fisher matrix clip value | $10^{-3}$ |
| | SI Fisher matrix multiplier | $10^{-6}$ |
| | nr. epochs | 4 |
| 1st experience | lr $\phi$ (feature extractor) | $3 \cdot 10^{-4}$ |
| | lr $\psi$ (classification head) | $3 \cdot 10^{-4}$ |
| | nr. epochs | 4 |
| Following experiences | lr $\phi$ (feature extractor) | $3 \cdot 10^{-4}$ |
| | lr $\psi$ (classification head) | $3 \cdot 10^{-4}$ |

Table 4: Hyper-parameters of the model trained with **no replay**. Common hyper-parameters are the same for each experience, 1st experience hyper-parameters are used in the first experience, the following experience hyper-parameters are used in all the following experiences.

| | Hyper-parameter | Value |
|---|---|---|
| | optimizer | SGD |
| | momentum | 0.9 |
| Common | weight decay | $10^{-4}$ |
| | minibatch size | 128 |
| | SI (Synaptic Intelligence) | disabled |
| | nr. epochs | 4 |
| 1st experience | lr $\phi$ (feature extractor) | $3 \cdot 10^{-2}$ |
| | lr $\psi$ (classification head) | $3 \cdot 10^{-2}$ |
| | nr. epochs | 4 |
| | lr $\phi$ (feature extractor) | $5 \cdot 10^{-5}$ |
| | lr $\psi$ (classification head) | $5 \cdot 10^{-4}$ |
| Following experiences | memory size | 1,500 |
| | replay pattern in the minibatch | 14 |
| | latent replay layer | conv5_4 |

Table 5: Hyper-parameters of the model trained with **replay** (generative replay, random data, and real data). Common hyper-parameters are the same for each experience, 1st experience hyper-parameters are used in the first experience, the following experience hyper-parameters are used in all the following experiences.

## F.2 IMAGENET-1000 NC

|  | Hyper-parameter | Value |
|---|---|---|
| **Common** | optimizer | SGD |
|  | momentum | 0.9 |
|  | weight decay | $10^{-4}$ |
|  | minibatch size | 128 |
|  | SI (Synaptic Intelligence) | disabled |
| **1st experience** | nr. epochs | 45 |
|  | lr $\phi$ (feature extractor) | $10^{-1}$ |
|  | lr $\psi$ (classification head) | $10^{-1}$ |
|  | lr scheduler | lr $\cdot$ 0.1 every 15 epochs |
| **Following experiences** | nr. epochs | 32 |
|  | lr $\phi$ (feature extractor) | $5 \cdot 10^{-3}$ |
|  | lr $\psi$ (classification head) | $5 \cdot 10^{-2}$ |
|  | lr scheduler | see Equation 16 |

Table 6: Hyper-parameters of the model trained with **no replay**. Common hyper-parameters are the same for each experience, 1st experience hyper-parameters are used in the first experience, the following experience hyper-parameters are used in all the following experiences.

|  | Hyper-parameter | Value |
|---|---|---|
| **Common** | optimizer | SGD |
|  | momentum | 0.9 |
|  | weight decay | $10^{-4}$ |
|  | minibatch size | 128 |
|  | SI (Synaptic Intelligence) | disabled |
| **1st experience** | nr. epochs | 45 |
|  | lr $\phi$ (feature extractor) | $10^{-1}$ |
|  | lr $\psi$ (classification head) | $10^{-1}$ |
|  | lr scheduler | lr $\cdot$ 0.1 every 15 epochs |
| **Following experiences** | nr. epochs | 32 |
|  | lr $\phi$ (feature extractor) | $5 \cdot 10^{-3}$ |
|  | lr $\psi$ (classification head) | $5 \cdot 10^{-2}$ |
|  | lr scheduler | see Equation 16 |
|  | memory size | 20,000 |
|  | replay pattern in the minibatch | 36 |
|  | latent replay layer | layer4 (4th resnet block) |

Table 7: Hyper-parameters of the model trained with **replay** (generative replay and real data). Common hyper-parameters are the same for each experience, 1st experience hyper-parameters are used in the first experience, the following experience hyper-parameters are used in all the following experiences.

Due to the complexity of the ImageNet-1000 scenario, we found it useful to use a learning rate scheduler that decreases the learning rate as the number of experiences progresses. The scheduler can be formalized as:

$$\text{lr} = \text{lr}_{init} \cdot \left( -\frac{0.9}{1 + e^{-1.5i + 8}} + 1 \right), \tag{16}$$

where $i$ indicates the index of the current experience.

## F.3   CORE50 NIC

| | Hyper-parameter | Value |
|---|---|---|
| Common | optimizer | SGD |
| | momentum | 0.9 |
| | weight decay | $10^{-4}$ |
| | minibatch size | 128 |
| | SI (Synaptic Intelligence) $\lambda$ | $2.3 \cdot 10^6$ |
| | SI Fisher matrix clip value | $10^{-3}$ |
| | SI Fisher matrix multiplier | $2 \cdot 10^{-5}$ |
| 1st experience | nr. epochs | 4 |
| | lr $\phi$ (feature extractor) | $10^{-3}$ |
| | lr $\psi$ (classification head) | $10^{-3}$ |
| Following experiences | nr. epochs | 4 |
| | lr $\phi$ (feature extractor) | $10^{-4}$ |
| | lr $\psi$ (classification head) | $10^{-3}$ |

Table 8: Hyper-parameters of the model trained with **no replay**. Common hyper-parameters are the same for each experience, 1st experience hyper-parameters are used in the first experience, the following experience hyper-parameters are used in all the following experiences.

| | Hyper-parameter | Value |
|---|---|---|
| Common | optimizer | SGD |
| | momentum | 0.9 |
| | weight decay | $10^{-4}$ |
| | minibatch size | 128 |
| | SI (Synaptic Intelligence) | disabled |
| 1st experience | nr. epochs | 4 |
| | lr $\phi$ (feature extractor) | $10^{-3}$ |
| | lr $\psi$ (classification head) | $10^{-3}$ |
| Following experiences | nr. epochs | 4 |
| | lr $\phi$ (feature extractor) | $10^{-4}$ |
| | lr $\psi$ (classification head) | $10^{-3}$ |
| | memory size | 300 (N/A for random data) |
| | replay pattern in the minibatch | 64 (21 for random data) |
| | latent replay layer | conv5_4 |

Table 9: Hyper-parameters of the model trained with **replay** (generative replay, random data, and real data). Common hyper-parameters are the same for each experience, 1st experience hyper-parameters are used in the first experience, the following experience hyper-parameters are used in all the following experiences.

### F.4   ON THE AMOUNT OF REPLAY DATA IN THE MINIBATCH

The amount of replay data included in each minibatch has a direct impact on the performance of the continual learning strategy adopted. We observed that the optimal value changes with the quality of the replay data, and that a large amount of degraded replay data in each minibatch may decrease disruptively the performance of the model.

We compared different original/replay proportions, finding that when using real replay data, the model is not much sensitive to the amount of replay data in the minibatch and different proportions work well: we empirically noticed a peak of performance around a 50-50 split. Using generated (degraded) or random data is quite different. We noticed that if the data used is highly degraded the

maximum gain in performance is when 10-30% replay data are added. Exceeding 30% usually leads to a degradation of performance, and if the amount of replay data is still higher (depending on the replay data quality) the accuracy of the model can be lower than not using replay data.

# G  GENERATIVE MODEL HYPER-PARAMETERS

## G.1  CORE50 NC

|  | Hyper-parameter | Value |
|---|---|---|
|  | optimizer | Adam |
|  | betas | 0.9 - 0.999 |
|  | weight decay | 0 |
|  | minibatch size | 128 |
|  | latent space dim. | 100 |
| Common | $\beta$ | 0.1 |
|  | $\eta$ | 0.01 |
|  | lr | $2 \cdot 10^{-3}$ |
|  | lr scheduler | None |
|  | nr. epochs | 4 |
| Following experiences | replay patterns in the minibatch | 27 |

Table 10: Hyper-parameters of the generative model trained on CORe50 NC. Common hyper-parameters are the same for each experience, while following experience hyper-parameters are used in all the experiences except the first one.

## G.2  IMAGENET-1000 NC

|  | Hyper-parameter | Value |
|---|---|---|
|  | optimizer | SGD |
|  | momentum | 0 |
|  | weight decay | 0 |
|  | minibatch size | 128 |
|  | latent space dim. | 100 |
| Common | $\beta$ | 0.25 |
|  | $\eta$ | 0.01 |
|  | lr | 1 |
|  | lr scheduler | see Equation 16 |
|  | nr. epochs | 32 |
| Following experiences | replay patterns in the minibatch | 36 |

Table 11: Hyper-parameters of the generative model trained on ImageNet-1000 NC. Common hyper-parameters are the same for each experience, while following experience hyper-parameters are used in all the experiences except the first one.

### G.3 CORE50 NIC

|  | Hyper-parameter | Value |
|---|---|---|
|  | optimizer | Adam |
|  | betas | 0.9 - 0.999 |
|  | weight decay | 0 |
|  | minibatch size | 128 |
| Common | latent space dim. | 100 |
|  | $\beta$ | 0.1 |
|  | $\eta$ | 0.01 |
|  | lr | $2 \cdot 10^{-3}$ |
|  | lr scheduler | None |
|  | nr. epochs | 4 |
| Following experiences | replay patterns in the minibatch | 64 |

Table 12: Hyper-parameters of the generative model trained on CORe50 NIC. Common hyper-parameters are the same for each experience, while following experience hyper-parameters are used in all the experiences except the first one.

## H ADDITIONAL PLOTS

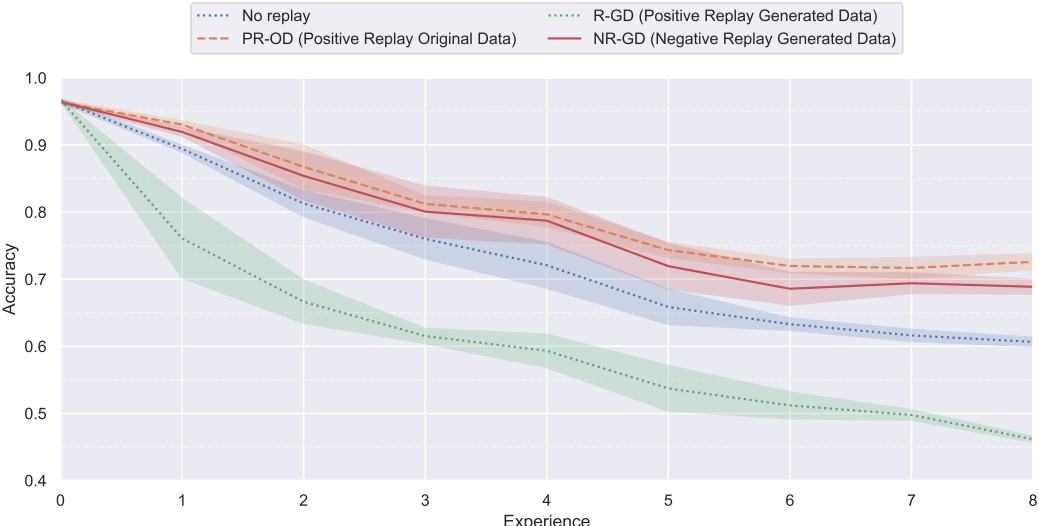

Figure 8: Overall accuracy on the CORe50 NC scenario, using a growing test set. After each experience, the model was evaluated using a test composed of only data belonging to the classes seen so far, similar to the benchmark proposed by Masana et al. (2020). Every experiment is averaged over 3 runs, with different seeds and class order. The standard deviation is reported in light colors. Better viewed if zoomed on a computer monitor.

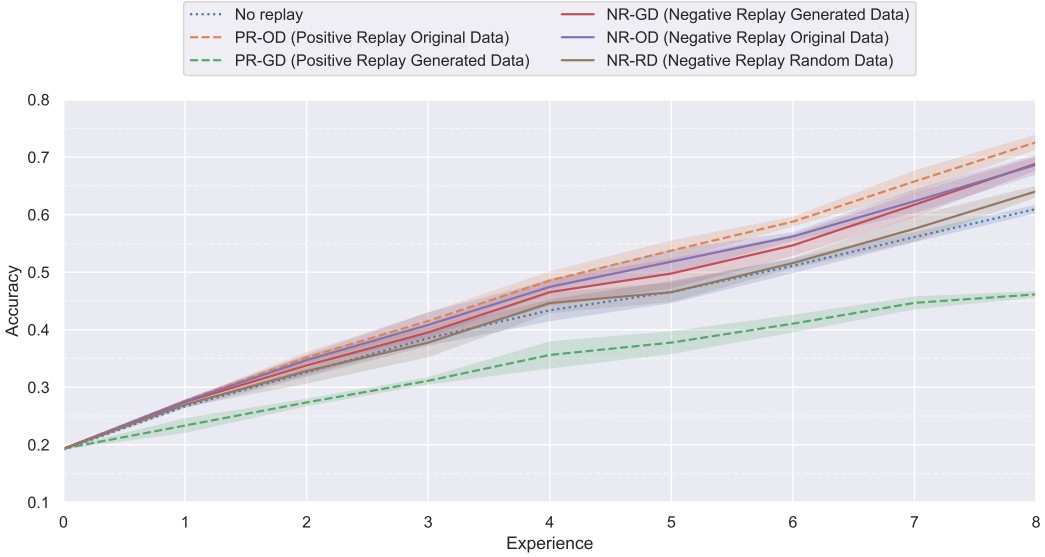

Figure 9: Overall accuracy on the CORe50 NC scenario for all the experiments performed in this work (included random data and negative replay with original data). After each experience, the model was evaluated using the cumulative test set as proposed by Lomonaco & Maltoni (2017). Every experiment is averaged over 3 runs, with different seeds and class order. The standard deviation is reported in light colors. better viewed if zoomed on a computer monitor.

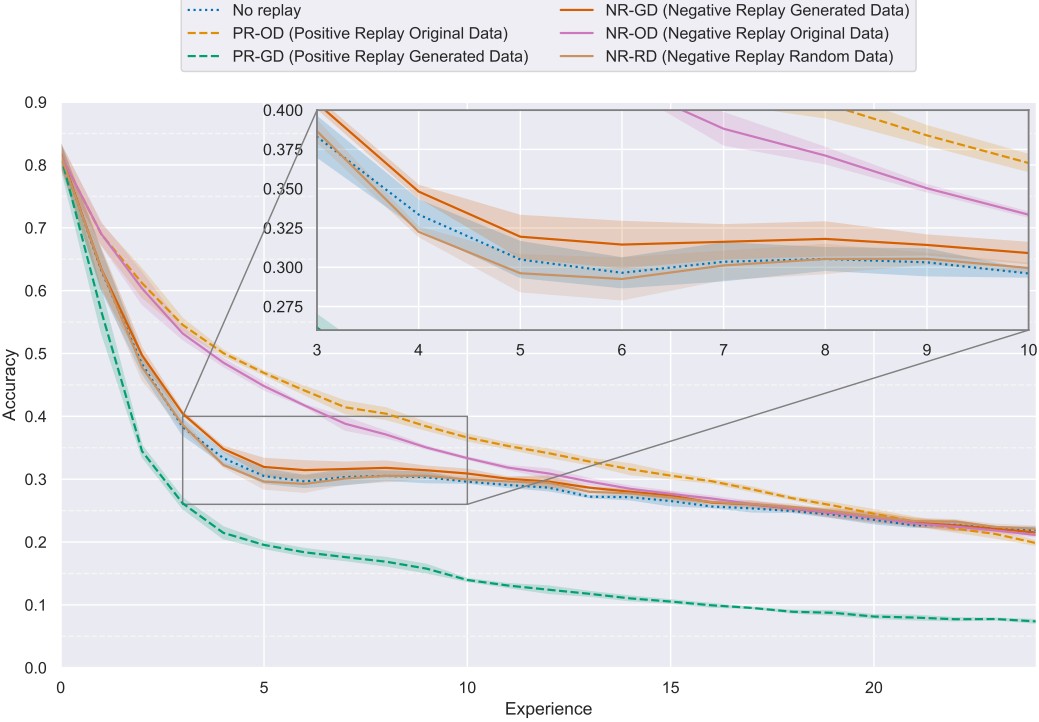

Figure 10: Overall accuracy on the ImageNet-1000 NC scenario for all the experiments performed in this work (included random data and negative replay with original data). After each experience, the model was evaluated using the whole test set as proposed by Masana et al. (2020). Every experiment is averaged over 3 runs, with different seeds and class order. The standard deviation is reported in light colors. better viewed if zoomed on a computer monitor.

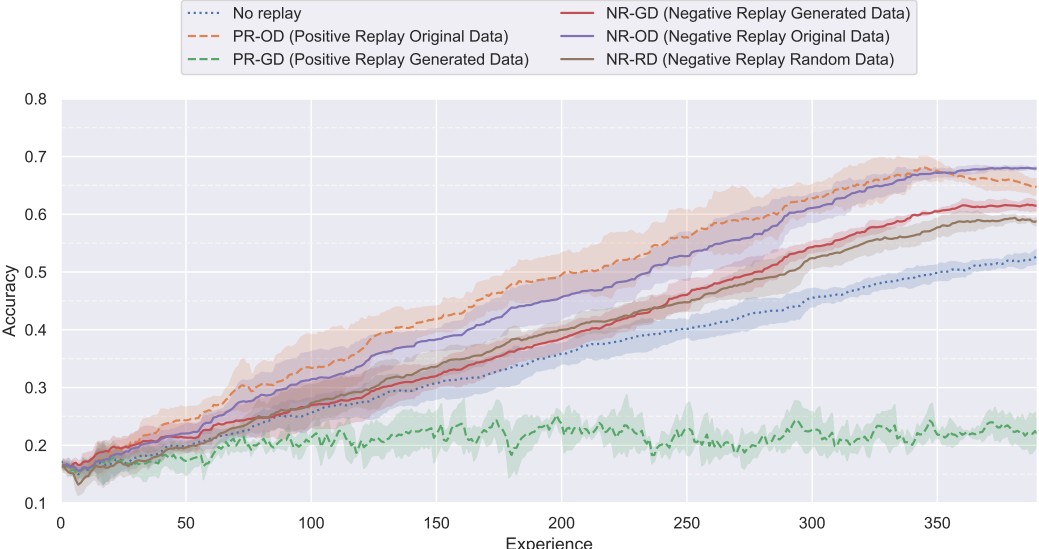

Figure 11: Overall accuracy on the CORe50 NIC scenario for all the experiments performed in this work (included random data and negative replay with original data). After each experience, the model was evaluated using the cumulative test set as proposed by Lomonaco & Maltoni (2017). Every experiment is averaged over 3 runs, with different seeds and class order. The standard deviation is reported in light colors. better viewed if zoomed on a computer monitor.

