# OpenReview forum: "Generative Negative Replay for Continual Learning"
_ICLR.cc/2022/Conference — ICLR 2022 Submitted_

### Official Review · Reviewer_mn5p · 2021-10-27

**Correctness:** 2
**Technical Novelty And Significance:** 3
**Empirical Novelty And Significance:** 1
**Recommendation:** 3
**Confidence:** 5

**Main Review:**


The paper presents an interesting method that could improve how to use replay in continual learning.
There is a high amount of experiments with various baselines.
Unfortunately, the baselines are not run to estimate the interest of the replay strategy but to select only one model that will use the replay strategy.
It would make more sense to run the different baseline approach with and without the replay strategy to estimate the interest of the proposed method.

The authors beat the state of the art in ImageNet1000 and Core50 with a combination of their method and a model inspired by several papers.
Beating state of the art is good; however, it does not provide a good evaluation of the replay strategy.
The sota experiments would be interesting as a conclusion experiment after evaluating negative replay on several baseline strategies.


Masking past output neurons while applying backward pass for generative replay is new, however, it has already been experimented in continual learning papers in other contexts such as "Continual Learning in Deep Networks: an Analysis of the Last Layer" Lesort et al.
The related works should also mention how the proposed masking strategy is related to existing works.


Other Remarks:

- the approach is designed to make low-quality replayed data useful. However, no assessment of the replay quality is presented. The authors compare the results of replay with generated data, real data, and random data without actually evaluating the data quality.
- the approach for Core50 and Imagenet is not the same (one pretrained feature extractor and one feature extractor trained only on task 1), making it difficult to assess the results.
- the paper should be clear from the beginning that the replay process is realized in the latent space.
- in conclusion : "using negative replay largely improves" is maybe a bit over enthusiastic (+2.95% for Core50, +0.83% Imagenet)
- Table 3: The accuracy reported from [Masana et al] is not EWC but from EWC-E, which is a version of EWC with Replay.
- Figure 2: The mask should appear more clearly and annotate into the figure. Why the backward pass of new samples goes to past classes? New data does not bring any information about past data.
- The related works section presents a bibliography of generative replay but do not provide perspective on the link between the presented approach and the bibliography. The context described in the related works section should be linked to the paper approach.

**Summary Of The Paper:**

The paper proposes a method to make generative replay for continual more effective even if generated data quality is not perfect. The method uses replayed data only as negative samples for current tasks and not as training samples to remember.
They apply their method on one model on Core50 and ImageNet data.
The replay process is realized in the latent space.

**Summary Of The Review:**

Negative Replay looks like a promising approach to continual learning with approximate generative models. The papers propose interesting results with several baselines.
However, the experiments are not designed in a convincing setup to evaluate negative replay. The evaluation of negative replay would benefit to be applied on several classical approaches before being applied on a specific/complex training procedure.

---

> ### Author Response · Authors · 2021-11-20
> **Authors response to Reviewer mn5p - part 1**
>
> Thank you for your insightful and constructive comments, which help improve the completeness of our submission. We uploaded an updated version of the paper with the modifications required.
>
> We have taken all the comments into consideration and summarized the responses below:
>
> > The paper presents an interesting method that could improve how to use replay in continual learning. There is a high amount of experiments with various baselines. Unfortunately, the baselines are not run to estimate the interest of the replay strategy but to select only one model that will use the replay strategy. It would make more sense to run the different baseline approach with and without the replay strategy to estimate the interest of the proposed method.
>
> We agree that evaluating the generative negative replay idea in combination with other baseline CL approaches could be interesting, and this is one of the next goals for us. However, the integration is often not straightforward, and running experiments on the complex dataset we adopted can require weeks for a single experiment (in fact, according to our re-implementation experience AR1 is one of the fastest techniques).
>
> To prove the efficacy of generative negative replay independently of the strategy we have included in the updated version of the paper (Appendix C) a preliminary test where negative replay is used without AR1. In particular, we tested the negative replay idea in conjunction with the “naive” CL approach (i.e., fine-tuning throughout the experiences without any mechanism to contrast forgetting). The implementation was done by masking the gradient as illustrated in Figure 2 and Figure 5. As expected performances are lower than with AR1, but the relative ranking of the replay methods (PR-GD, No replay, NR-GD, PR-OD) is maintained.
>
> We report the table with the final accuracy (averaged over 3 different runs) here:
>
> | __Naive__         | __Naive PR-GD__  | __Naive PR-OD__  | __Naive NR-GD__      |
> |-------------------|------------------|------------------|----------------------|
> | 22.17 $\pm$ 1.00  | 25.55 $\pm$ 2.86 | 67.60 $\pm$ 2.39 | __36.88 $\pm$ 1.74__ |
>
> > The authors beat the state of the art in ImageNet1000 and Core50 with a combination of their method and a model inspired by several papers. Beating state of the art is good; however, it does not provide a good evaluation of the replay strategy. The sota experiments would be interesting as a conclusion experiment after evaluating negative replay on several baseline strategies.
>
> The goal of this paper is to prove that generative negative replay can be a valid alternative to generative replay in the complex setting where training incrementally a generator is still too challenging. Exploring the full potential of negative replay in combination with other state-of-the-art approaches could be better investigated in future work by us and other research groups.
>
> > Masking past output neurons while applying backward pass for generative replay is new, however, it has already been experimented in continual learning papers in other contexts such as "Continual Learning in Deep Networks: an Analysis of the Last Layer" Lesort et al. The related works should also mention how the proposed masking strategy is related to existing works.
>
> Thank you for pointing out this interesting paper, we missed it since it was released as a preprint on arXiv only a few months ago. We agree that the masking concept is used in other works, but we only use it as an example of how negative replay can be applied independently of a continual learning strategy. In the experiments, we used AR1, which uses CWR [3] to control the forgetting on the last layer. In this case, the masking is not necessary since a slight modification of CWR is sufficient to implement negative replay. Nevertheless, in the new version of the paper, we have included more pointers to the related literature.
>
> > the approach is designed to make low-quality replayed data useful. However, no assessment of the replay quality is presented. The authors compare the results of replay with generated data, real data, and random data without actually evaluating the data quality.
>
> We agree that quantitatively studying how the replay data quality influences the performance of the model (both positive replay and negative replay) could be interesting. However, since we are generating samples in the latent space such evaluation is not simple: classical metrics (SSIM [4], PSNR to the Inception score [5] or the Fréchet inception distance [6]) are not much useful and in high-dimensional spaces (features have thousands of dimensions) the concept of distance may be misleading [7]. So we will reconsider this issue in our future studies.

---

> > ### Author Response · Authors · 2021-11-20
> > **Authors response to Reviewer mn5p - part 2**
> >
> > > the approach for Core50 and Imagenet is not the same (one pretrained feature extractor and one feature extractor trained only on task 1), making it difficult to assess the results.
> >
> > We followed the benchmark rules proposed in the original papers, [3] for core50 ([8] for the NIC benchmark) and [1] for ImageNet-1000). We agree that this may cause some confusion, but using different datasets with different starting models (pretrained and not pretrained) is not new, since many publications in the past used different approaches when dealing with different benchmarks (e.g. [2] use a model pretrained on CIFAR-10 on the experiments on CIFAR-100, but no pre-training is used in the experiments with MNIST).
> >
> > > the paper should be clear from the beginning that the replay process is realized in the latent space.
> >
> > We have modified our paper to make this clear from the beginning.
> >
> > > in conclusion : "using negative replay largely improves" is maybe a bit over enthusiastic (+2.95% for Core50, +0.83% Imagenet)
> >
> > With “largely improve” we meant w.r.t. the traditional “positive” generative replay approach, where we obtain an improvement of ~11.5% on CORe50 NC, ~14.5% on ImageNet-1000 NC, and about 40% on the CORe50 NIC. Actually, on NIC the improvement is significant also with respect to  the no replay baseline (+10%). We have changed the sentence in the conclusions to clarify that large improvements refer to positive replay.
> >
> > > Table 3: The accuracy reported from [Masana et al] is not EWC but from EWC-E, which is a version of EWC with Replay.
> >
> > Thank you for pointing this out. We edited the name in the updated version of the paper.
> >
> > > Figure 2: The mask should appear more clearly and annotate into the figure. Why the backward pass of new samples goes to past classes? New data does not bring any information about past data.
> >
> > In NIC scenario (unlike NC) the new samples (called original in the figure) can belong to already encountered classes and need to be back propagated to old classes. We have updated the figure caption in the updated version of the paper, to better explain the point. We also included another figure in Appendix C (Figure 6) that better explain the concept.
> >
> > > The related works section presents a bibliography of generative replay but do not provide perspective on the link between the presented approach and the bibliography. The context described in the related works section should be linked to the paper approach.
> >
> > We improved the related work section which is now better linked to the proposed approach.
> >
> > ---
> >
> > ### References
> >
> > [1] Masana, M., Liu, X., Twardowski, B., Menta, M., Bagdanov, A. D., & van de Weijer, J. (2020). Class-incremental learning: survey and performance evaluation on image classification. arXiv preprint arXiv:2010.15277.
> >
> > [2] van de Ven, G. M., Siegelmann, H. T., & Tolias, A. S. (2020). Brain-inspired replay for continual learning with artificial neural networks. Nature communications, 11(1), 1-14.
> >
> > [3] Maltoni, D., & Lomonaco, V. (2019). Continuous learning in single-incremental-task scenarios. Neural Networks, 116, 56-73.
> >
> > [4] Wang, Z., Bovik, A. C., Sheikh, H. R., & Simoncelli, E. P. (2004). Image quality assessment: from error visibility to structural similarity. IEEE transactions on image processing, 13(4), 600-612.
> >
> > [5] Salimans, T., Goodfellow, I., Zaremba, W., Cheung, V., Radford, A., & Chen, X. (2016). Improved techniques for training gans. Advances in neural information processing systems, 29, 2234-2242.
> >
> > [6] Heusel, M., Ramsauer, H., Unterthiner, T., Nessler, B., & Hochreiter, S. (2017). Gans trained by a two time-scale update rule converge to a local Nash equilibrium. Advances in neural information processing systems, 30.
> >
> > [7] Aggarwal, C. C., Hinneburg, A., & Keim, D. A. (2001, January). On the surprising behavior of distance metrics in high dimensional space. In the International conference on database theory (pp. 420-434). Springer, Berlin, Heidelberg.
> >
> > [8] Lomonaco, V., Maltoni, D., & Pellegrini, L. (2020, June). Rehearsal-Free Continual Learning over Small Non-IID Batches. In CVPR Workshops (pp. 989-998).

---

> > > ### Comment · Reviewer_mn5p · 2021-11-29
> > > **Answer**
> > >
> > > Thanks for the detailed answer to my review.
> > > The authors commented on most of my comments, they added an experience in the appendix to compare negative replay without negative replay on CORe50 NC - 8  tasks and made some modification in the text to be more clear.
> > > I still believe the methodology of this paper needs to improve a lot, and they should focus on a better evaluation of their strategy.
> > > I understand that they wanted to show that negative replay *can* outperform normal replay, but I expect more insight and analysis for a research paper.
> > > Therefore I maintain my recommendation.

---

### Official Review · Reviewer_9MUi · 2021-11-01

**Correctness:** 3
**Technical Novelty And Significance:** 2
**Empirical Novelty And Significance:** 2
**Recommendation:** 5
**Confidence:** 4

**Main Review:**

Strengths:
1. The paper is well-written and states the problem setting quite well.
2. Using generated examples as negative examples is interesting and novel (for generative replay based continual learning methods).

Weakness:
1. The main method is based on AR1, is there a particular reason of using this AR1? Could the method still work well under the case that only generative replay is allowed (no additional components as in AR1)?
2. To my understanding, the main purpose of the paper is to learn new data better, instead of reducing forgetting on the old data. However, according to Figure 2 (c), the gradient will also flows back to the feature extractor (when it is not frozen). This may leads to forgetting issue, when no other techniques (Synaptic Intelligence or Experience Replay using real data) are applied.
3. There are actually many tricks for learning a better classifier in the continual learning literature, see Labels Trick [1], Separated Softmax [2], etc., which also works without any replay techniques. It would be idea to compare with those methods.
4. (Minor) The novelty and practical usage of the method is limited by the fact that a generative model is required.

[1] Zeno et al. Task agnostic continual learning using online variational bayes. Arxiv 2018.
[2] Ahn et al.  A simple class decision balancing for incremental learning. Arxiv 2020.

**Summary Of The Paper:**

This paper proposes a novel learning scheme for generative replay methods in continual learning. Different from the original generative replay method, the generated examples are treated as negative examples for new classes. Experimental results on CORe50 and ImageNet-1000 demonstrate the effectiveness of the proposed method.

**Summary Of The Review:**

The idea is interesting, but the novelty is limited, and some details and comparison experiments are missing.

---

> ### Author Response · Authors · 2021-11-20
> **Authors response to Reviewer 9MUi**
>
> Thank you for your insightful and constructive comments, which help improve the completeness of our submission. We uploaded an updated version of the paper with the modifications required.
>
> We have taken all the comments into consideration and summarized the responses below:
>
> > The main method is based on AR1, is there a particular reason for using this AR1? Could the method still work well under the case that only generative replay is allowed (no additional components as in AR1)?
>
> This question is very relevant. In the paper we focused on AR1 because::
> - it allows a simple and efficient implementation of the negative replay idea (mainly due to CWR).
> - it is competitive with other state-of-the-art approaches (see Appendix D).
> - it natively supports latent replay [1].
> However, we recognize that our claim that generative negative replay is effective independently of the strategy remains unproven. Validating negative replay without AR1 or alongside other continual learning strategies is one of the most important future works.
>
> In the new version of the paper (Appendix C), some preliminary experiments are reported to support this claim. In particular, we tested the negative replay idea in conjunction with the “naive” CL approach (i.e., fine-tuning throughout the experiences without any mechanism to contrast forgetting). The implementation was done by masking the gradient as illustrated in Figure 5. As expected performances are lower than with AR1, but the relative ranking of the replay methods (PR-GD, No replay, NR-GD, PR-OD) is maintained.
>
> We reported the table with the final accuracy (averaged over 3 different runs) here:
>
>  | __Naive__         | __Naive PR-GD__  | __Naive PR-OD__  | __Naive NR-GD__      |
> |-------------------|------------------|------------------|----------------------|
> | 22.17 $\pm$ 1.00  | 25.55 $\pm$ 2.86 | 67.60 $\pm$ 2.39 | __36.88 $\pm$ 1.74__ |
>
> > To my understanding, the main purpose of the paper is to learn new data better, instead of reducing forgetting on the old data. However, according to Figure 2 (c), the gradient will also flows back to the feature extractor (when it is not frozen). This may leads to forgetting issue, when no other techniques (Synaptic Intelligence or Experience Replay using real data) are applied.
>
> You are right. We also were worried about this, so in the first implementation(s) we also selectively blocked the gradient flow through the feature extractor, but no significant changes were observed, so we opted for simplicity. We believe that the feature extractor is less sensitive to forgetting, especially if a pre-trained model is used. Moreover, replay data, even if degraded, may still contain some structure that resembles real data, allowing the feature extractor to forget slowly in comparison to the classification head. We added some comments in the new version of the paper.
>
> > There are actually many tricks for learning a better classifier in the continual learning literature, see Labels Trick [1], Separated Softmax [2], etc., which also works without any replay techniques. It would be idea to compare with those methods.
>
> We agree that there are many strategies or tricks that can work without replay, and we will consider them in the future for comparisons. However, the main objective of this work was comparing generative replay with generative negative replay.
>
> > (Minor) The novelty and practical usage of the method is limited by the fact that a generative model is required.
>
> This is true, but we explicitly address the problem of generative replay in this work, so a generative model is required by definition. However, as also pointed out by reviewer onET, this method could also work without a generative model, using data from other sources (e.g., random crops) as negative examples. Our experiments in section 4.4, where we used random data as negative examples (without any generative model), proved that even totally unrealistic data can provide some benefit.

---

### Official Review · Reviewer_onET · 2021-11-01

**Correctness:** 3
**Technical Novelty And Significance:** 2
**Empirical Novelty And Significance:** 3
**Recommendation:** 5
**Confidence:** 4

**Main Review:**

Strengths:
- Using generated data as only negative examples is interesting. It reminds me of SVM Examples [1], where they train N binary classifiers (where N is the amount of data). To train each classifier, this method uses only 1 positive example and the others as negative examples. In Exemplar SVM, the model learns very well what not to classify. I understand that the idea of Generated Negative Examples is different, but maybe it could be helpful.
- I find it an ingenious idea to use CWR to train the sorting head. The model only focuses on making controlled changes, ignoring changes that can generate unwanted noise.
- I liked how the paper is structured, introducing the current problem of methods that use generated inputs. Then, how this can be alleviated by using negative replay as negative examples.

Comments and Questions to the authors:
- Acknowledging that one of the problems for not using data from past experiences is data privacy. Instead of generating negative examples, you could use the same proposed technique but actual data (extracted from other parts without security problems). Since you only train the columns of the classifier of current experience, there is no supervision needed. Maybe this brings the results a little closer to the real examples, but without security problems?
- On page 5, just below Eq 7. What do you mean by “patterns” is similar to a data point?
- In Figure 4, why do you think that after the 350 experience, the accuracy of the PR-OD starts to drop, but using NR-GD it remains stable? Do you think it could be due to the poor representation of the data from past experiences during the training? Is this drop due to the number of items from past experiences in the batch?
- How does the proportion of elements generated in the batch affect it? I imagine that if you place fewer elements, the performance drops, but how sensitive is it to this number?
- I understand that this method only works when we have little data in each experience. This is since there would not be enough data to correctly train the columns of the classifier of the classes that are not present in the experience. Could this technique be used in other scenarios? Not necessarily when we have few classes from experience. Perhaps using techniques similar to ACL to train the adversarial model?

Typos:
- Table 1 and 2, said NR-OG. I think it should have said NR-OD

[1] Malisiewicz, Tomasz, Abhinav Gupta, and Alexei A. Efros. "Ensemble of exemplar-svms for object detection and beyond." 2011 International conference on computer vision. IEEE, 2011.

**Summary Of The Paper:**

In Continual Learning (CL) scenarios, storing data from previous experience is helpful to mitigate catastrophic forgetting. However, privacy issues or storage overhead makes replay methods impractical. Aware of this problem, generative models have been proposed in previous work to generate data that represent previous experiences. Still, these methods suffer from low performance because of the continual training of the generating model and/or the generation of high dimensionality data. In this paper, the authors propose Generative Negative Replay. Instead of using the generated data as a positive example of previous classes, they used it as a negative example for the classes present in the current experience. While training with the negative examples, the authors propose to freeze the weights of the classification head corresponding to the previous classes, similar to what happens in CWR. This solution is particularly relevant in situations where there are not as many classes from experience.

**Summary Of The Review:**

The authors do not present a new idea but a very ingenious solution for a recurring problem. Having little data to train the tasks does not impede training well if you have a little imagination. By generating data that cannot be used directly as positive examples (because of the limitations of generation methods), they use this data as negative examples for other tasks. Although I still doubt if this can be extended to other scenarios, for example, when tasks are more similar or different. I think the authors present surprising results.

---

> ### Author Response · Authors · 2021-11-20
> **Authors response to Reviewer onET - part 1**
>
> Thank you for your insightful and constructive comments, which help improve the completeness of our submission. We uploaded an updated version of the paper with the modifications required.
>
> We have taken all the comments into consideration and summarized the responses below:
>
> > Using generated data as only negative examples is interesting. It reminds me of SVM Examples [1], where they train N binary classifiers (where N is the amount of data). To train each classifier, this method uses only 1 positive example and the others as negative examples. In Exemplar SVM, the model learns very well what not to classify. I understand that the idea of Generated Negative Examples is different, but maybe it could be helpful.
>
> Thank you for pointing that paper out; the general idea is similar since both use data that does not belong to any class (in the case of one class SVM all the data from the other classes, in the case of this paper the generative data that is highly degraded and do not represent the objects in the training set) to improve the classification performance. We will check this paper and the related literature, since interesting ideas from possible future works may be inspired by this field of research.
>
> > Acknowledging that one of the problems for not using data from past experiences is data privacy. Instead of generating negative examples, you could use the same proposed technique but actual data (extracted from other parts without security problems). Since you only train the columns of the classifier of current experience, there is no supervision needed. Maybe this brings the results a little closer to the real examples, but without security problems?
>
> When designing the experiments, we thought of using data from another dataset instead of generated data, or random crops from the same dataset (without including the object to be classified). All these methods have the advantage of not requiring a generative model, and the data can easily be extracted from the current experience. We will surely investigate this path in the future, since we believe it can be very promising, given that even using random data showed a noticeable increase in accuracy.
>
> > On page 5, just below Eq 7. What do you mean by “patterns” is similar to a data point?
>
> Yes, we changed “pattern” to “data point” in the updated version of the paper to avoid confusion.
>
> > In Figure 4, why do you think that after the 350 experience, the accuracy of the PR-OD starts to drop, but using NR-GD it remains stable? Do you think it could be due to the poor representation of the data from past experiences during the training? Is this drop due to the number of items from past experiences in the batch?
>
> The distribution of class patterns throughout NIC 391 experiences can lead to a sort of saturation in the last 30-40 experiences when all the classes have been already introduced, and only new instances of existing classes are provided (see also [ 1] ). PR-OD probably reached maximum accuracy around experience 350 and some overfitting could have occurred later. On the other hand, NR-GD accuracy at 350 is lower so it has more margin to learn and, during the last 40 experiences, the trend is maintained.
>
> > How does the proportion of elements generated in the batch affect it? I imagine that if you place fewer elements, the performance drops, but how sensitive is it to this number?
>
> This is an interesting point. Actually, we did several tests on this but they are not included in the paper for lack of space. We noted that the sensitivity to the amount of replay data in the minibatch depends on the quality of the replay samples. When using real replay data, the model is not much sensitive to the amount of replay data in the minibatch and different proportions work well: we empirically noticed a peak of performance around a 50-50 split.
>
> Using generated (degraded) or random data is quite different. We noticed that if the data used is highly degraded the maximum gain in performance is when 10-30% replay data are added. Exceeding 30% usually leads to a degradation of performance, and if the amount of replay data is still higher (depending on the replay data quality) the accuracy of the model can be lower than not using replay data. We added some comments on this to the new version of the paper (see appendix F.4).

---

> > ### Author Response · Authors · 2021-11-20
> > **Authors response to Reviewer onET - part 2**
> >
> > > I understand that this method only works when we have little data in each experience. This is since there would not be enough data to correctly train the columns of the classifier of the classes that are not present in the experience. Could this technique be used in other scenarios? Not necessarily when we have few classes from experience. Perhaps using techniques similar to ACL to train the adversarial model?
> >
> > Yes, we believe that the proposed technique can be used in other scenarios, mainly in weakly supervised settings or to address open set classification. For example in (some) weakly supervised scenarios, we could consider unlabeled data as negative examples. Oviously, this must be done with caution to avoid model drifts due to data mislabelling, but there are applications where extra constraints (e.g., temporal and spatial coherence) can be exploited to control the risk.
> >
> > > Table 1 and 2, said NR-OG. I think it should have said NR-OD
> >
> > Thank you, corrected in the updated version.
> >
> > ---
> >
> > ### References:
> > [1] Pellegrini, L., Graffieti, G., Lomonaco, V., & Maltoni, D. (2020, March). Latent replay for real-time continual learning. In 2020 IEEE/RSJ International Conference on Intelligent Robots and Systems (IROS) (pp. 10203-10209). IEEE.

---

> > ### Comment · Reviewer_onET · 2021-11-29
> > **Thanks for the response**
> >
> > This work presents a very interesting line of research since it takes advantage of data that otherwise would not be a contribution to the training process. However, I have doubts about how and why the method really works. On the one hand, they manage to train without forgetting, but this is mainly due to the contribution of AR1 / CWR, on the other hand, they achieve better accuracy, but I do not know if this is actually due to the use of generated data or if we can use any type of data (which is the main focus of the paper). I think this work is on a very good path, but it still lacks a bit before it can be accepted at this conference.

---

### Official Review · Reviewer_SNYr · 2021-11-04

**Correctness:** 3
**Technical Novelty And Significance:** 3
**Empirical Novelty And Significance:** Not applicable
**Recommendation:** 6
**Confidence:** 4

**Main Review:**

### **Strengths**
- The key idea of using generative patterns/samples only as negative samples is simple, yet effective. The weight updates by the generated imperfect samples can be derogatory for the classes learnt in the past experiences. But these samples are still important for the new classes being learnt in the current experience due to the "learning in isolation problem".

- Empirically, this approach is found to significantly outperform both positive generative replay and no-replay paradigms. Also, when compared to the PR-OD (Positive replay with original dataset), which is an upper bound for continual learning with replay approaches, the generative negative replay comes reasonably close in performance

- The datasets used in this paper such as ImageNet-1000 indicate that this approach could be scalable to more complex or real-world scenarios of continual learning for classification.

- The paper is generally well written and is an easy read.

### **Weaknesses** and **Suggestions**
- One of my concerns with that approach is that splitting the model into classification and feature extraction heads adds another hyperparameter. How does one decide where the split will be? Readers might want an answer to this question before using this approach.
- It is a little surprising to see the accuracy on the ImageNet-1000 decreasing with the number of experiences, while the accuracy of CORe50 increases with the number of experiences. Is it because they are calculated differently? Maybe for CORe50, the accuracy is calculated across all classes even though they haven't been trained yet, while this is not the case for ImageNet-1000.
  - If yes, it would be better to use the same way to calculate the accuracy, otherwise it is quite confusing
  - If no, then it would be interesting to understand why this is happening?

- Is PR-OD the same as Cumulative scores (Pelligrino et. al. 2020)? One of the goals of continual learning is to learn models (that are incrementally exposed to all the training data) that are comparable to a model that was trained on the full dataset. Hence, instead of PR-OD cumulative scores would probably be a better upper bound.
- For experiments on CORe50, the MobileNetV1 was first pre-trained on ImageNet-1000 (to probably get a good feature extractor). But what about experiments on ImageNet1000; was the ResNet-18 pretrained? If yes, on which dataset?
  - Another question that could be raised here is the choice of pretrained model as the feature extractor. Does it matter how the pre-trained model was trained? For example, if an unsupervised/self-supervised approach is used that uses negative examples and learns linearly separable embeddings, will it reduce the need for generative negative replay for continual learning?
- The definition of the generative model $g_\Omega$ is slightly confusing. In eq 5, the input to $g_\omega$ is $z$, while in equation 6 the input now becomes a sample from the input space $\mathcal{X}$.

### Typos
- Table 1: PR-OD instead of PR-OG


**Summary Of The Paper:**

The key idea proposed in this paper is to use generative replay as negative samples to improve performance in class-incremental scenarios of continual learning. The authors argue that samples from a generative model have a lot of artefacts due to challenges in training/adapting a low resource generative model. Hence, its use as a positive sample for future experiences fails. But, these imperfect samples can still be relied on as negative samples for classes being trained in the current experience to prevent "learning in isolation" problems. The proposed approach is compared to positive replay, negative replay and no-replay baselines for two complex datasets in both NC and NIC scenarios.

**Summary Of The Review:**

Overall the key idea proposed in the paper is sound and reasonably well justified with experiments. That being said, I have some concerns regarding some of the design choices and is the approach robust to these design choices. Hence, I have given a borderline rating.

---

> ### Author Response · Authors · 2021-11-20
> **Authors response to Reviewer SNYr - part 1**
>
> Thank you for your insightful and constructive comments, which help improve the completeness of our submission. We uploaded an updated version of the paper with the modifications required.
>
> We have taken all the comments into consideration and summarized the responses below:
>
> > One of my concerns with that approach is that splitting the model into classification and feature extraction heads adds another hyperparameter. How does one decide where the split will be? Readers might want an answer to this question before using this approach.
>
> Thank you for pointing this out. As discussed in [1] the choice of the latent replay layer has an impact on the tradeoff accuracy-efficiency but it is not critical, since the model behavior for different choices changes smoothly. In this paper, in the experiment with Core50, we chose the same latent replay layer as in [1], while for ImageNet-1000 we chose the 4th block of the resnet-18 network, mainly for efficiency. We clarified this issue in the updated version of the paper.
>
> > It is a little surprising to see the accuracy on the ImageNet-1000 decreasing with the number of experiences, while the accuracy of CORe50 increases with the number of experiences. Is it because they are calculated differently? Maybe for CORe50, the accuracy is calculated across all classes even though they haven't been trained yet, while this is not the case for ImageNet-1000.
> > - If yes, it would be better to use the same way to calculate the accuracy, otherwise it is quite confusing
> > - If no, then it would be interesting to understand why this is happening?
>
> You are right, this is quite confusing. However, we followed the “default” reporting mode of the existing benchmarks (on the full test set as proposed in [2] for Core50 NC and with a growing test set as proposed in [3] for ImageNet-1000). This was briefly explained in the captions of Figures 3 and 4, but we agree that the reader can find this disturbing. Therefore, in the updated version of the paper, we included the plots on Core50 using the growing test set as in the experiments on ImageNet-1000, making possible a direct comparison.
>
> We prefer the fixed test set approach used in CORe50  [2] since the difficulty of the problem remains the same, and one can better appreciate if the model is learning (or not) throughout the experiences. In the growing test set case, the problem becomes more difficult as the training progresses (more classes) and even if the model is learning the trend can be negative.
>
> > Is PR-OD the same as Cumulative scores (Pellegrini et. al. 2020)? One of the goals of continual learning is to learn models (that are incrementally exposed to all the training data) that are comparable to a model that was trained on the full dataset. Hence, instead of PR-OD cumulative scores would probably be a better upper bound.
>
> In (Pellegrini et al. 2020) the cumulative strategy trains the model on all the data seen so far. The PR-OD strategy in this paper is not the same as the cumulative, since only a limited number of data points from past experiences are used for training (i.e., those in the Replay memory).
>
> Our PR-OD can be conceived as an upper bound to generative replay, since a generative model can, at its maximum capacity, produce data that is indistinguishable from real data, so the performance of the model using real replay data is expected to be the best case.
>
> The cumulative, on the other hand, is expected to be the upper bound for any continual learning strategy. For CORe50, using a MobileNetV1 (as in the paper) the cumulative approach reached an accuracy of about 85%, while for ImageNet-1000, using a ResNet-18, the cumulative approach reached an accuracy of about 70%.
>
> > For experiments on CORe50, the MobileNetV1 was first pre-trained on ImageNet-1000 (to probably get a good feature extractor). But what about experiments on ImageNet1000; was the ResNet-18 pretrained? If yes, on which dataset?
>
> The ResNet-18 used on ImageNet-1000 was not pretrained, because this is the “default” for this benchmark and we provide a comparison with existing approaches in Appendix D, Table 3 . We pointed out this in the updated version of the paper.

---

> > ### Author Response · Authors · 2021-11-20
> > **Authors response to Reviewer SNYr - part 2**
> >
> > > Another question that could be raised here is the choice of pretrained model as the feature extractor. Does it matter how the pre-trained model was trained? For example, if an unsupervised/self-supervised approach is used that uses negative examples and learns linearly separable embeddings, will it reduce the need for generative negative replay for continual learning?
> >
> > This is a very interesting question, and it deserves a thorough and independent study. We agree that using negative examples in the pre-training stage could lead to a more robust representation learning, but at this stage, we cannot guess the resulting impact on the learning in isolation problem. Another interesting research direction could be inserting noise during the pre-training, simulating the degraded data that can come from a generative model. In this case, the model might become more robust to degraded data that do not belong to any class or even mislabeled data. We have included these considerations in future works in the updated version of the paper.
> >
> > > The definition of the generative model $g_\Omega$ is slightly confusing. In eq 5, the input to $g_\Omega$ is $z$, while in equation 6 the input now becomes a sample from the input space $\mathcal{X}$.
> >
> > Thank you for pointing this out. The two definitions may be misleading at the first sight. In equation 5 we consider a generic generative model, which takes as input a random vector and conditioning information, and outputs a generated data point from the learned data distribution. In equation 6 we suppose to have a generative model composed of an encoder and a decoder (see the paragraph before equation 6). In this case, the encoder $q_\gamma$ takes as input a data point from the training set ($x \in \mathcal{X}$) and outputs a latent vector $z$, which is then taken as input by the encoder $p_\xi$ to produce the generated data. We have modified the paragraph before equation 6 to make the distinction more clear.
> >
> > > Table 1: PR-OD instead of PR-OG
> >
> > Thank you, corrected in the updated version.
> >
> > ---
> >
> > ### References
> > [1] Pellegrini, L., Graffieti, G., Lomonaco, V., & Maltoni, D. (2020, March). Latent replay for real-time continual learning. In 2020 IEEE/RSJ International Conference on Intelligent Robots and Systems (IROS) (pp. 10203-10209). IEEE.
> >
> > [2] Maltoni, D., & Lomonaco, V. (2019). Continuous learning in single-incremental-task scenarios. Neural Networks, 116, 56-73.
> >
> > [3] Masana, M., Liu, X., Twardowski, B., Menta, M., Bagdanov, A. D., & van de Weijer, J. (2020). Class-incremental learning: survey and performance evaluation on image classification. arXiv preprint arXiv:2010.15277.

---

### Decision · Program_Chairs · 2022-01-20

**Decision:**

Reject

**Comment:**

This paper suggests a new technique to utilize generative replay for continual learning. Specifically, the authors claim that even though the generated samples are imperfect (thus cannot be used as positive samples for old classes), they can still be used as negative samples for the current class. 3 reviewers are negative and 1 reviewer is positive. The main concerns of negative reviewers are (a) non-ablated effects of baseline and proposed components, (b) insufficient analysis of negative replay, and (c) no assessment of generated data quality. The rebuttal provides an additional experiment to address the issue (a), but the reviewers and AC think the experiments should be better polished. Also, AC believes the issues (b) and (c) should be better analyzed. The rebuttal claims that issue (c) is not applicable as they generate samples on the latent space. However, the main motivation of the paper is the low quality of generated samples, and the paper should provide a quality measure to support their claim. For example, an update of the feature extractor may move the latent space generative replay to the wrong class (i.e., low quality), and thus one should not use it as positive but only as negative, as suggested in this paper. Here, the negative replay would increase the margin of current and old classes, enhancing the accuracy of the current class. To analyze the source of benefits (old vs. current classes), the authors could report the task-wise accuracy trends, not only the overall accuracy. It would be a nice addition to the issue (b). Due to these unresolved concerns, AC tends to recommend rejection.